# Unravelling cysteine-deficiency-associated rapid weight loss

Alan Varghese[1,2,12], Ivan Gusarov[2,12], Begoña Gamallo-Lana[3], Daria Dolgonos[1], Yatin Mankan[1], Ilya Shamovsky[2], Mydia Phan[1], Rebecca Jones[4], Maria Gomez-Jenkins[5,6], Eileen White[5,6,7], Rui Wang[8], Drew R. Jones[2], Thales Papagiannakopoulos[9], Michael E. Pacold[10], Adam C. Mar[3], Dan R. Littman[1,11✉] & Evgeny Nudler[2,11✉]

Around 40% of the US population and 1 in 6 individuals worldwide have obesity, with the incidence surging globally[1,2]. Various dietary interventions, including carbohydrate, fat and, more recently, amino acid restriction, have been explored to combat this epidemic[3–6]. Here we investigated the impact of removing individual amino acids on the weight profiles of mice. We show that conditional cysteine restriction resulted in the most substantial weight loss when compared to essential amino acid restriction, amounting to 30% within 1 week, which was readily reversed. We found that cysteine deficiency activated the integrated stress response and oxidative stress response, which amplify each other, leading to the induction of GDF15 and FGF21, partly explaining the phenotype[7–9]. Notably, we observed lower levels of tissue coenzyme A (CoA), which has been considered to be extremely stable[10], resulting in reduced mitochondrial functionality and metabolic rewiring. This results in energetically inefficient anaerobic glycolysis and defective tricarboxylic acid cycle, with sustained urinary excretion of pyruvate, orotate, citrate, α-ketoglutarate, nitrogen-rich compounds and amino acids. In summary, our investigation reveals that cysteine restriction, by depleting GSH and CoA, exerts a maximal impact on weight loss, metabolism and stress signalling compared with other amino acid restrictions. These findings suggest strategies for addressing a range of metabolic diseases and the growing obesity crisis.

The pioneering work of William C. Rose in 1937 revealed nine essential amino acids (EAAs): histidine, isoleucine, leucine, lysine, methionine, phenylalanine, threonine, tryptophan and valine[11]. Notably, cysteine is also essential in animals with mutations in either cystathionine γ-lyase (CSE, also known as CTH and CGL) or cystathionine β-synthase (CBS), enzymes of the *trans*-sulfuration pathway[12,13] (Fig. 1a). Extensive research has examined the effects of removing individual EAAs, shedding light on their roles in metabolism, energy expenditure and weight and fat loss[14–18].

Amino acid deprivation triggers the integrated stress response (ISR) through GCN2, which detects uncharged tRNAs and phosphorylates translation initiation factor eIF2α[19]. Phosphorylated eIF2α suppresses general translation while promoting translation of the key ISR transcription factor ATF4 and its downstream targets, including FGF21 and GDF15[20–22].

The sulfur amino acid restriction (SAAR) diet, that is, a diet of combined low methionine and cysteine (Cys), is notable because it increases lifespan and protects against metabolic diseases in rodents

and nematodes[23–26]. However, it is unclear whether the benefits of SAAR are driven by methionine or cysteine restriction.

Cysteine is not only a proteinogenic amino acid but is also the limiting intermediary metabolite in glutathione (GSH) biosynthesis[12,13,27] (Fig. 1a). Cysteine also has a critical, although underappreciated, role alongside pantothenic acid (vitamin B5, hereafter B5) in CoA synthesis[28–30]. CoA is also considered to be extremely stable, as mice on B5-deficient diets for as long as 2 months do not show significant loss of CoA[10].

The growing interest in diets that induce weight loss prompted us to compare restriction of each individual EAA and cysteine. Our findings revealed that cysteine deficiency induces the most weight loss compared with all other EAAs, resulting in a 30% reduction of body weight within 7 days. Our experiments have elucidated a coordinated mechanism underlying this phenomenon, characterized by the rapid induction of ISR and oxidative stress response (OSR), accompanied by increased GDF15 and FGF21, and a reduction in CoA levels resulting in metabolic inefficiency, therefore offering insights for potential intervention in metabolic diseases and body-weight control.

[1]Department of Cell Biology, NYU Grossman School of Medicine, New York, NY, USA. [2]Department of Biochemistry and Molecular Pharmacology, NYU Grossman School of Medicine, New York, NY, USA. [3]Department of Neuroscience and Physiology, Neuroscience Institute, NYU Grossman School of Medicine, New York, NY, USA. [4]Division of Advanced Research Technologies, NYU Grossman School of Medicine, New York, NY, USA. [5]Rutgers Cancer Institute, Rutgers University, New Brunswick, NJ, USA. [6]Department of Molecular Biology and Biochemistry, Rutgers University, Piscataway, NJ, USA. [7]Ludwig Princeton Branch, Ludwig Institute for Cancer Research, Princeton University, Princeton, NJ, USA. [8]Department of Biology, York University, Toronto, Ontario, Canada. [9]Department of Pathology, Laura and Isaac Perlmutter Cancer Center, NYU Langone Health, New York, NY, USA. [10]Department of Radiation Oncology and Laura and Isaac Perlmutter Cancer Center, NYU Langone Health, New York, NY, USA. [11]Howard Hughes Medical Institute, New York, NY, USA. [12]These authors contributed equally: Alan Varghese, Ivan Gusarov. ✉e-mail: Dan.Littman@med.nyu.edu; Evgeny.Nudler@nyulangone.org

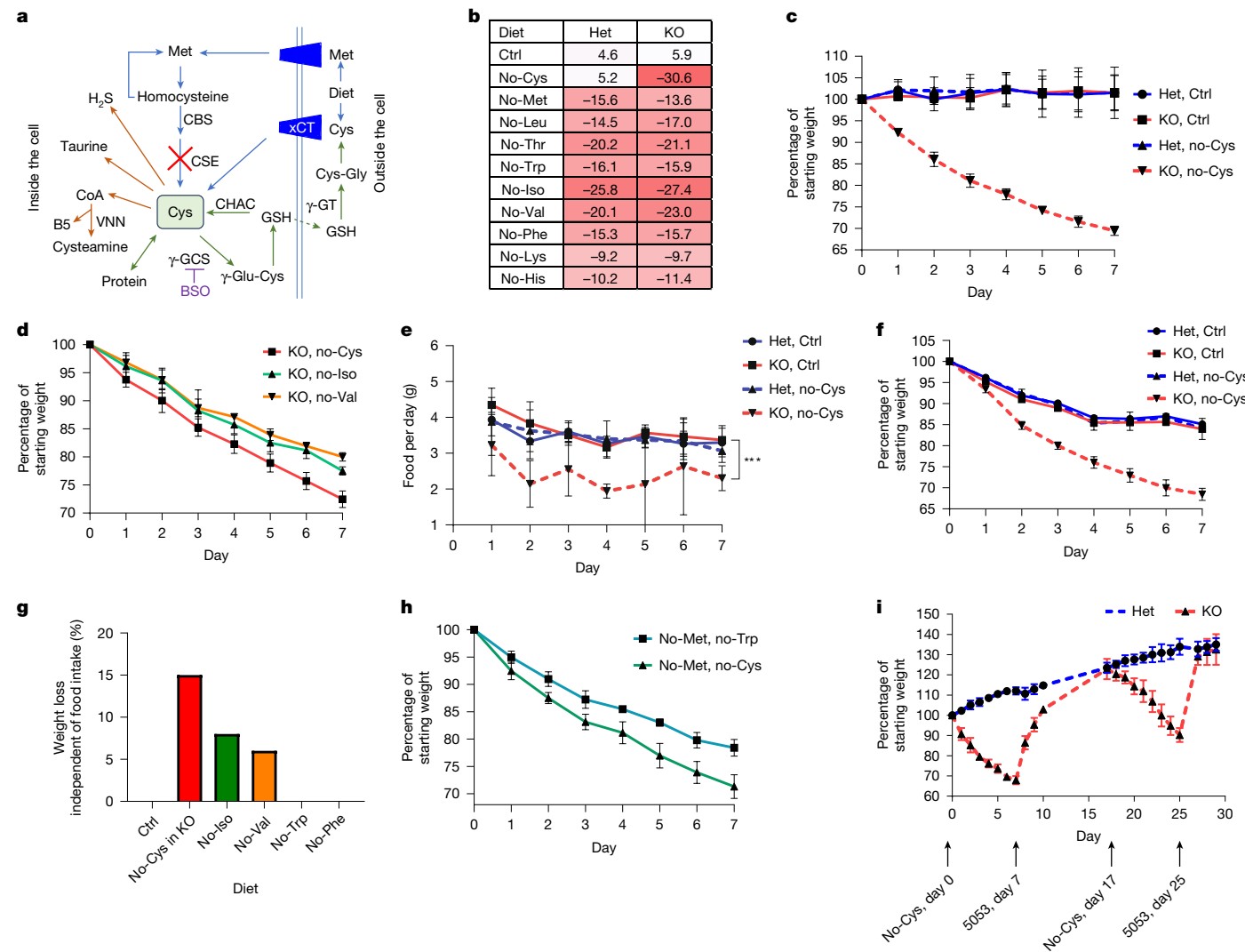

**Fig. 1 | Cysteine deficiency drives rapid weight loss. a**, A simplified cartoon demonstrating the pathways for Cys synthesis (blue) and consumption (reversible, green; and irreversible, brown). CSE deletion is marked by the red cross. The GSH synthesis inhibitor BSO is shown in purple. VNN, vanin/pantetheinase; GT, glutamyl transferase; GCS, glutamate–cysteine ligase. **b**, The average percentage of weight loss relative to the starting weight after removal of each EAA and cysteine in male $Cse^{+/-}$ (Het) and $Cse^{-/-}$ (KO) mice for 1 week. $n = 5$ ($Cse^{-/-}$ control and no-Cys) and $n = 4$ (other groups). **c**, Daily weight curves of male $Cse^{+/-}$ and $Cse^{-/-}$ mice fed control (Ctrl) or no-Cys diets, house at 22 °C. $n = 4$ per group. **d**, Daily weight curves of $Cse$-KO mice deprived of isoleucine, valine or cysteine at 30 °C. $n = 4$ per group. **e**, Daily food consumption of $Cse^{+/-}$ or $Cse^{-/-}$ mice on control or no-Cys diets. $n = 3$ per group. **f**, Body-weight curves of CR (2.1 g per day) male $Cse^{+/-}$ and $Cse^{-/-}$ mice on the control 5CC7 and no-Cys diets. $n = 4$ per group. **g**, The average percentage weight loss unaccounted for by reduced food consumption (including data for Iso and Val and for Trp and Phe)[14,18]. **h**, The weight of male C57BL/6 (B6) mice on CR with no-Met no-Trp compared to no-Met no-Cys. $n = 4$ per group. **i**, Weight curves of male $Cse^{+/-}$ or $Cse^{-/-}$ mice over cycles of no-Cys diet versus standard chow 5053. $n = 4$ per group. Statistical analysis was performed using repeated-measures one-way analysis of variance (ANOVA, **e**). Data are mean ± s.d. ***$P < 0.001$.

## Cys deprivation induces rapid weight loss

We evaluated the weight loss induced when each of the EAAs and cysteine were individually removed from diet in both $Cse$ knockout (KO; $Cse^{-/-}$) and $Cse$ heterozygous ($Cse^{+/-}$) C57BL/6 mice. Cysteine deprivation in $Cse^{-/-}$ mice, but not in heterozygous and wild-type (WT) mice, led to the largest weight loss compared with other EAAs[14] (Fig. 1b,c and Extended Data Fig. 1a–e). A cysteine-free (no-Cys) diet induced weight loss exclusively in $Cse^{-/-}$ mice, indicating that depletion of newly absorbed and synthesized cysteine is necessary for the effect (Extended Data Fig. 1f). We also observed consistent percentage weight loss across different starting weights (Extended Data Fig. 1g–j). Female mice displayed slightly lower weight loss on day 1, a difference that remained constant (Extended Data Fig. 1k).

Weight loss was completely prevented by supplying cysteine through either *N*-acetylcysteine (NAC) or GSH (which is broken down to cysteine

in the gut)[31] (Extended Data Fig. 1l). There were no differences in amino acids, vitamins, glucose or palmitic acid in the stool between $Cse^{+/-}$ and $Cse^{-/-}$ mice, indicating no defects in nutrient absorption (Extended Data Fig. 1m–o and Supplementary Table 1). Restoration of $H_2S$, a degradation product of cysteine, did not prevent weight loss[31] (Extended Data Fig. 1p). Microbiota alterations also did not explain weight loss, as antibiotic-treated $Cse^{-/-}$ mice and $Cse^{-/-}$ mice co-housed with $Cse^{+/-}$ mice had similar weight-loss profiles (Extended Data Fig. 1q). Considering the known roles of isoleucine and valine in thermogenesis in brown adipose tissue[32], we also measured weight loss in mice maintained at thermoneutral conditions (30 °C). Weight loss was still most pronounced for cysteine deficiency, with only a 2.7% change from the 22 °C condition (Fig. 1d).

Given that diets deficient in EAAs induce food aversion behaviour, we monitored the daily food consumption of $Cse^{-/-}$ and control heterozygous mice fed a no-Cys diet compared with a control diet[33]. $Cse^{-/-}$ mice

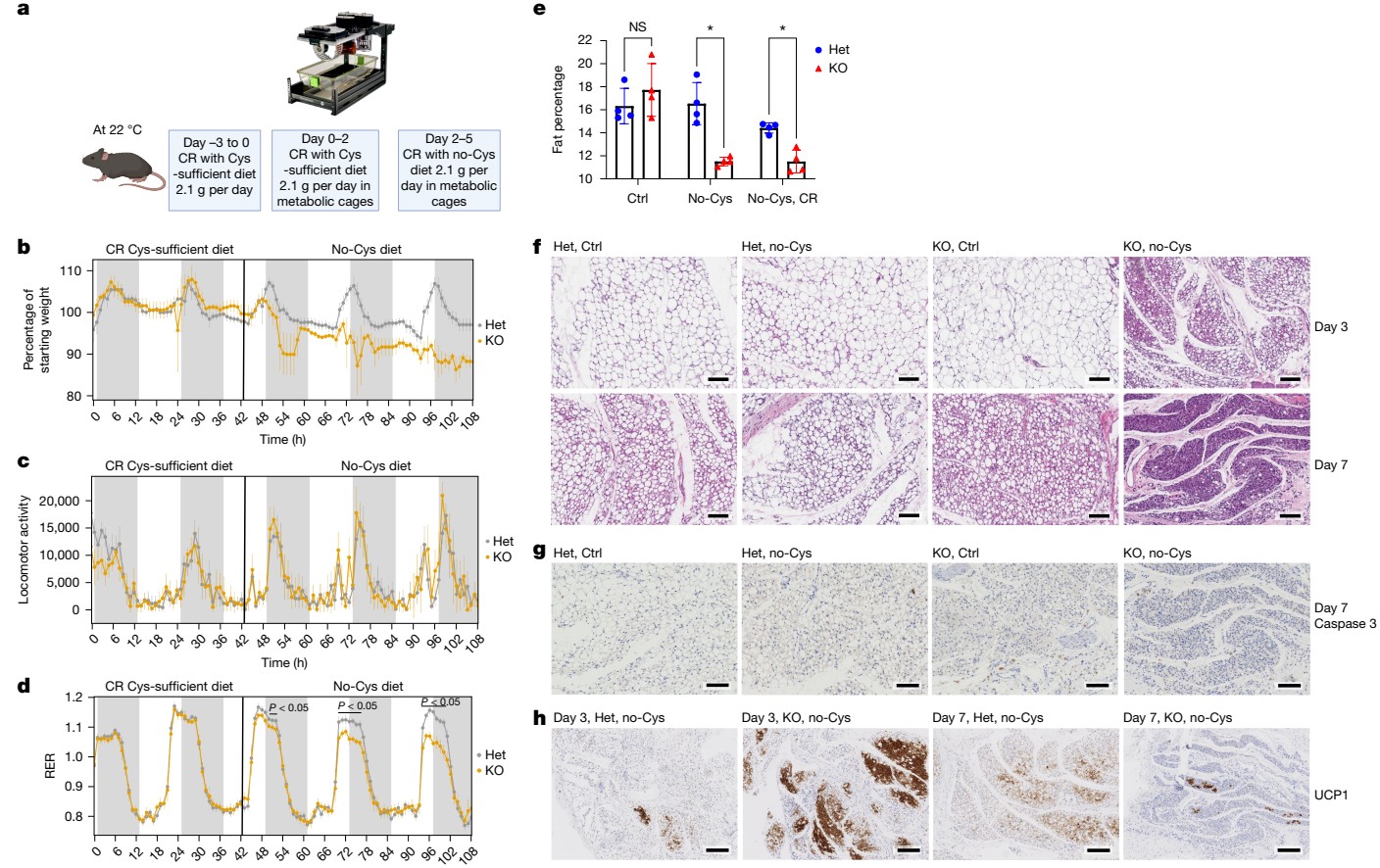

**Fig. 2 | Cysteine deficiency drives rapid metabolic changes and loss of white adipose tissue mass. a–d** Metabolic cage profiles of male *Cse*[+/–] and *Cse*[–/–] mice on CR (*n* = 5 (*Cse*[+/–]) and 3 (*Cse*[–/–])), demonstrating the experimental design (food was given daily between 2 and 3 pm) (**a**), body-weight measurements (**b**), locomotor activity (**c**) and the RER (**d**). **e**, Body fat percentages, as determined by DEXA scans on day 7, of mice fed on no-Cys and control diets, either CR or ad libitum. *n* = 4 per group. NS, not significant. **f–h**, Representative images (*n* = 4 per group) of subcutaneous fat pads from *Cse*[+/–] or *Cse*[–/–] male

mice on CR with Ctrl or no-Cys diets stained with haematoxylin and eosin (H&E) on days 3 and 7 of cysteine restriction (**f**), and by immunohistochemistry (IHC) staining of caspase 3 on day 7 of cysteine restriction (**g**) and of UCP1 on days 3 and 7 of cysteine restriction (**h**). The diagram in **a** was created using BioRender and image from https://www.tse-systems.com/. For **d** and **e**, statistical analysis was performed using multiple unpaired *t*-tests. Data are mean ± s.d. *$P < 0.05$. Scale bars, 100 μm (**f** and **g**) and 200 μm (**h**).

on the no-Cys diet exhibited a 30% reduction in daily food intake, from 3.5 g to 2.4 g per day (Fig. 1e), while no difference was observed in *Cse*[+/–] mice on the control or no-Cys diets (3.4 g in both). This food aversion, and the resultant caloric restriction (CR), could independently lead to rapid weight loss. However, while CR of 2.1 g per day led to a weight loss of only 15–16% in the control mice, the *Cse*[–/–] mice on the no-Cys diet experienced a substantial 31.5% weight loss within 1 week (Fig. 1f). Thus, at least 15% of the weight loss in *Cse*[–/–] mice could not be explained by reduced food intake alone. By contrast, for isoleucine and valine, the amount of weight loss unexplained by reduced food intake was 8% and 6%, respectively, as reported previously[14]. For other EAAs, such as tryptophan and phenylalanine, the entire weight loss was accounted for by reduced food intake[18] (Fig. 1g), further emphasizing the unique effect of cysteine deprivation.

To determine whether the weight loss was due to the accumulation of *trans*-sulfuration pathway intermediates[12], we provided *Cse*[–/–] mice with a diet lacking both methionine and cysteine and compared the effect to that of the no-Cys diet. The weight-loss patterns were identical for the two diets, suggesting that neither homocysteine nor cystathionine were contributing to the weight loss (Extended Data Fig. 1r). Moreover, WT mice on a CR diet (2.1 g per day) devoid of methionine and cysteine lost approximately 30% of their body weight within 1 week as compared to those on a diet devoid of methionine and tryptophan that lost only 20%

(Fig. 1h). This strongly suggests that the benefits of SAAR are primarily driven by cysteine limitation.

Notably, after mice on the no-Cys diet were reverted to a standard chow diet, they regained approximately two-thirds of the lost weight within 2 days and fully recovered within 4 days (Fig. 1i). When returned to the no-Cys diet, the mice promptly resumed losing weight at a similar rate, which was once again reversed immediately after reintroduction to the standard chow. This pattern highlights the high reversibility of cysteine-deprivation-induced weight loss without apparent detrimental effect.

## Selective fat burning and rapid browning

To further characterize the weight loss, we conducted metabolic and behavioural assessments of *Cse*[–/–] and *Cse*[+/–] mice. After adapting to CR with a control diet (2.1 g per day, given between 2 and 3 pm throughout the experiment), the mice were placed into metabolic cages and, 2 days later, were switched to the no-Cys CR diet (Fig. 2a). The shift to the no-Cys diet immediately triggered weight loss in the *Cse*[–/–] mice, with a 10% decrease over 3 days, compared with a 0% change for *Cse*[+/–] mice (Fig. 2b and Extended Data Fig. 2a). Notably, there were no significant differences in locomotion and movement between the *Cse*[–/–] and *Cse*[+/–] groups under any condition, indicating that the weight loss is not

attributable to increased physical activity in $Cse^{-/-}$ mice, and that they do not exhibit lethargy (Fig. 2c and Extended Data Fig. 2b).

The respiratory exchange ratio (RER), which shows whether mice are selectively burning fat or carbohydrate, progressively decreased from day 1 to day 3 in $Cse^{-/-}$ animals on a no-Cys diet, suggesting increased usage of fat as fuel (Fig. 2d and Extended Data Fig. 2c). DEXA scans revealed a substantial reduction in fat content in $Cse^{-/-}$ mice on day 7 of the no-Cys diet (Fig. 2e). No differences were observed between $Cse^{-/-}$ and $Cse^{+/-}$ mice when provided with the control diet.

Histological studies of white adipose tissue revealed that $Cse^{-/-}$ mice deprived of cysteine exhibited higher fat loss from individual adipocytes by day 3 and, by day 7, there was near complete depletion of fat content throughout the tissue (Fig. 2f). All three control groups maintained significant fat content within each adipocyte on day 3, with only a slight decrease by day 7. Caspase-3 staining on day 7 revealed that, despite substantial fat loss, there was no detectable adipocyte cell death (Fig. 2g).

However, by day 3 in $Cse^{-/-}$ mice on a no-Cys diet, a notable proportion of adipocytes that contained multiple small fat droplets instead of a single large droplet, resembling brown/beige adipose tissue. Immunostaining of the fat pad for UCP1 revealed robust browning of white adipose tissue, which on day 3 was faster and more pronounced than previously reported after 4 weeks of CR[34] (Fig. 2h). Moreover, there was a rapid loss of visceral fat in $Cse^{-/-}$ mice on a no-Cys diet (Extended Data Fig. 2d,e).

We did not observe significant differences in muscle (quadricep) histology across the four groups (Extended Data Fig. 2f,g). In the liver, there was no significant fat accumulation or apparent pathology (Extended Data Fig. 2h,i). These findings may explain the lack of any effects on movement and the ease with which $Cse^{-/-}$ mice on a no-Cys diet can recover.

## Transcriptional responses to Cys depletion

To gain insights into the molecular responses triggered by cysteine deprivation, we conducted bulk RNA-sequencing (RNA-seq) analysis of liver (which exhibits the highest expression of $Cse$), muscle (the most abundant tissue) and adipose tissue (the tissue most impacted by cysteine loss). To distinguish responses specific to cysteine deprivation, we included a group exposed to a tryptophan-deficient diet (Fig. 3a). After a 3-day CR control diet, $Cse^{-/-}$ and $Cse^{+/-}$ mice were shifted to either CR control, CR no-Cys or CR tryptophan-free (no-Trp) diets, and tissues were collected after 2 days.

Gene Ontology (GO) enrichment analysis of genes upregulated in the liver of $Cse^{-/-}$ mice on a no-Cys diet revealed prominent categories such as 'cellular response to xenobiotic stimulus', 'GSH' and 'small molecule' metabolic processes (Extended Data Fig. 3a and Supplementary Table 2). The latter category revealed strong upregulation of genes associated with ISR, including amino acid synthesis and one-carbon metabolism ($Mthfd2$, $Pycr1$, $Asns$, $Psat1$), tRNA charging, amino acid transporters ($Slc7a1$, $Slc1a4$, $Slc3a2$) and various stress-response genes ($Fgf21$, $Gdf15$, $Ddit3$, $Trib3$, $Atf5$, $Atf6$) (Fig. 3b and Extended Data Fig. 3b,c). Genes in the 'cellular response to xenobiotic stimulus' and 'glutathione metabolic processes' categories ($Nqo1$, $Gstm1$–$Gstm4$, $Gsta1$, $Gsta2$, $Srxn1$) are characteristic of NRF2-regulated OSR.

The liver has a central role in the metabolism of fatty acids (FA) and triglycerides (TGs), shifting between synthesis and breakdown in response to the fed and fasted states. Several GO categories related to cholesterol and lipid metabolism were also enriched (Fig. 3c and Extended Data Fig. 3a). Genes for sterol regulatory element-binding proteins (SREBPs), the master regulators of de novo lipogenesis ($Srebf1$) and cholesterol biosynthesis ($Srebf2$)[35], and other genes associated with cholesterol biosynthesis were significantly downregulated in $Cse^{-/-}$ mice on the no-Cys diet. Furthermore, genes associated with the increased import of lipid particles into the liver ($Vldlr$, $Lrp1l$, $Pcsk9$

and $Apoc1$) were also deregulated. This shift in gene expression suggests that, after cysteine restriction, the liver increases the import of very-low-density lipoproteins and low-density lipoproteins, while reducing endogenous lipogenesis[36–38] (Fig. 3c).

Although some genes that were typically associated with ISR and cholesterol and FA biosynthesis were also upregulated and downregulated, respectively, in $Cse^{-/-}$ mice on a no-Trp diet in comparison to $Cse^{+/-}$ and $Cse^{-/-}$ mice on the control diet, their induction on the no-Cys diet was significantly more pronounced, underscoring a special role of cysteine in the regulation of ISR and fat metabolism (Fig. 3b). This can be explained by a potential synergy between ISR (ATF4 signature) and OSR (NRF2 signature) in the liver (Fig. 3b and Extended Data Fig. 3a)—a phenomenon that was previously observed in cell lines and tumour models[39–41].

The transcriptional response to the no-Cys diet in muscle and liver tissue of $Cse^{-/-}$ mice showed only partial overlap. In muscle, genes involved in the 'response to oxidative stress' GO category were notably enriched. NRF2 (encoded by $Nfe2l2$) and its typical target genes (such as $Nqo1$, $Hmox1$ and $Gclc$) were significantly upregulated in response to cysteine deficiency, indicating a robust induction of OSR in muscle (Extended Data Fig. 3d,e and Supplementary Table 3). The induction of several genes related to the import and catabolism of branched-chain amino acids ($Slc7a2$, $Idh2$, $Ivd$, $Bcat2$) suggests that muscle uses less glucose as an energy source, instead relying on amino acids not used for translation. Notably, ISR was not upregulated in muscle, suggesting that the abundance of proteins in the tissue may prevent amino acid levels from falling below the threshold required to induce ISR at day 2 (Extended Data Fig. 3g).

In epididymal adipose tissue, canonical ISR or OSR signatures were not observed (Extended Data Fig. 3h and Supplementary Table 4). Significant suppression of SREBP1 ($Srebf1$) and its target genes, such as $Scd2$, $Acly$, $Acaca$ and $Pgd$, indicated a shutdown of de novo lipogenesis in adipose tissue (Fig. 3d,e). Many genes related to lipid metabolism exhibited increased expression, including thioesterases ($Acot1$–$4$), implying increased release of free FAs (FFAs) from adipocytes. Although adipose tissue is not the major site for ketone-body biosynthesis, genes in this pathway ($Acat3$ and $Hmgcs2$) were upregulated. Moreover, there was a mild increase in $Ucp1$ expression (Extended Data Fig. 3i).

## Role of ISR and OSR in weight loss

A cysteine-free diet can lead to a deficiency in GSH, which activates NRF2-dependent OSR[42,43]. Indeed, we detected a marked decrease in GSH within 2 days in the liver and muscle, but not in the subcutaneous adipose tissue, of $Cse^{-/-}$ mice (Fig. 4a and Extended Data Fig. 4a–c). $Chac1$ upregulation probably contributes to the decrease in GSH in the liver (Fig. 3b). This decline in GSH was accompanied by the nuclear localization of NRF2 and increased NRF2-regulated protein NQO1 by day 3 in the livers of cysteine-deprived $Cse^{-/-}$ mice, but not $Cse^{+/-}$ mice (Fig. 4b), demonstrating OSR activation.

Consistent with ISR activation, we detected an increase in phosphorylated eIF2α (p-eIF2α) in the livers of $Cse^{-/-}$ mice on the no-Cys diet compared with heterozygous mice on the no-Cys diet (Fig. 4c). To verify the ISR contribution to weight loss and food aversion, we attempted to create $Gcn2$ $Cse$ double-knockout mice. However, 80% of such mice displayed hindlimb paralysis and died with 8 weeks, therefore preventing us from proceeding further. This phenotype suggests that $Cse^{-/-}$ mice, even on standard chow, exhibit a basal ISR driven by $Gcn2$, as evidenced by the mild increase in ISR genes such as $Asns$ in the liver (Supplementary Table 2). Moreover, this experiment suggests that the ISR is absolutely required for adaptation to Cys deficiency.

As anticipated by the transcriptomics results, there was an increase in GDF15 and FGF21, stress hormones associated with ISR and OSR[44,45], in the serum of $Cse^{-/-}$ mice on the no-Cys diet, but not in $Cse^{+/-}$ mice or in mice fed other diets (Fig. 4d,e and Extended Data Figs. 3b,c and 4d,e). Moreover, both hormones were elevated in WT (B6) mice on a

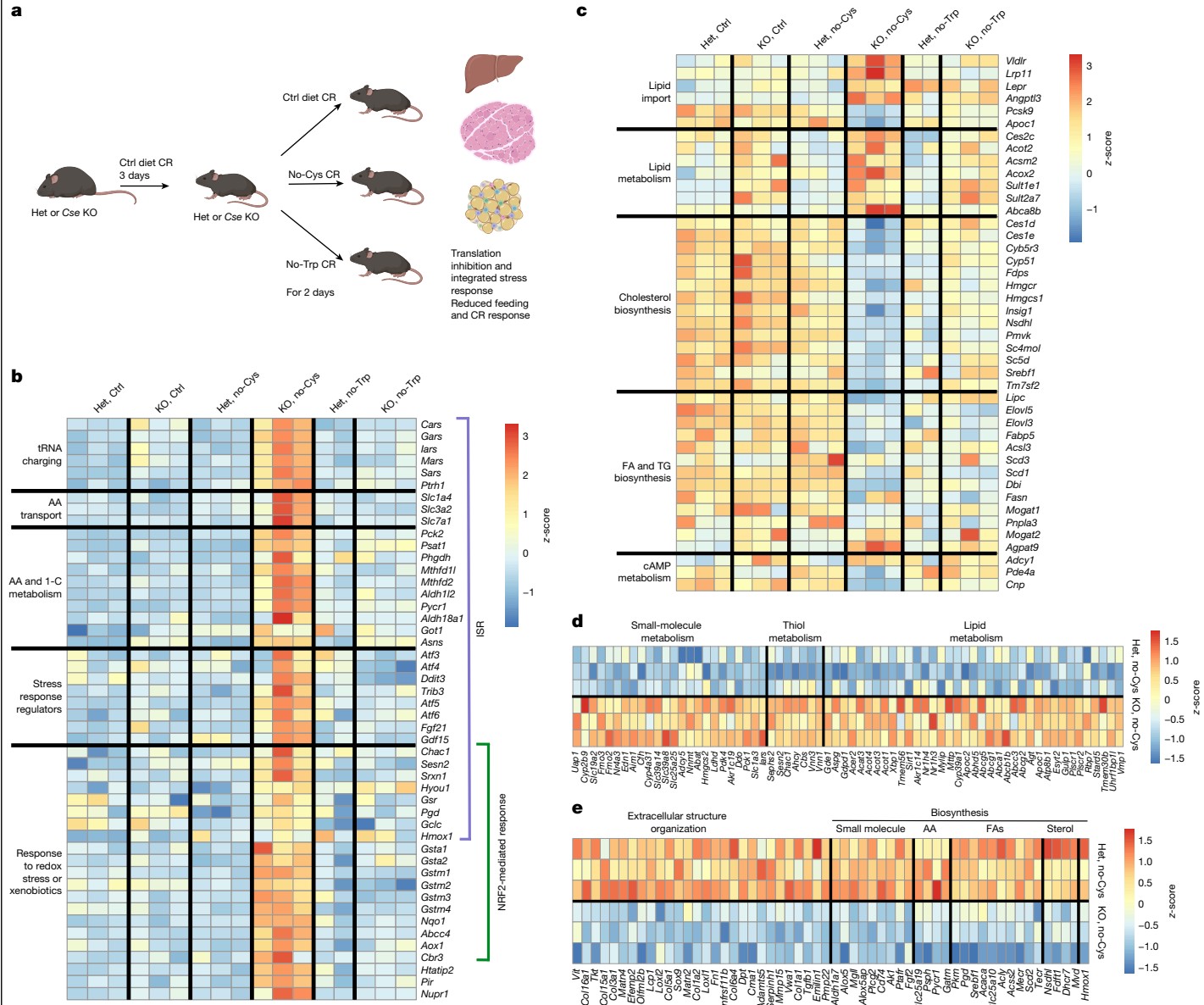

**Fig. 3 | Changes in gene expression induced by cysteine deficiency in liver and adipose tissues. a**, The experimental design for bulk mRNA-seq. **b,c**, Representative heat maps, illustrating the expression of genes related to ISR and OSR (**b**) and cholesterol and lipid synthesis and degradation (**c**) that are specifically upregulated or downregulated ($q < 0.05$) in the livers of KO mice on a no-Cys diet. AA, amino acids. **d,e**, Representative heat maps of genes that are specifically upregulated or downregulated in epididymal adipose tissue in KO no-Cys mice ($q < 0.05$), including genes related to lipid, thiol and

small-molecule metabolism (**d**), and extracellular organization, biosynthetic pathways in small molecules, amino acids, FAs and sterols (**e**). The diagram in **a** was created using BioRender. A complete list of differentially expressed genes for all sequenced samples is presented in Supplementary Tables 2 and 4. For the RNA-seq, the total numbers in each group were as follows: for liver, $Cse^{+/-}$ control (Ctrl) (5), $Cse^{-/-}$ control (4), $Cse^{+/-}$ no-Cys (8), $Cse^{-/-}$ no-Cys (7), $Cse^{+/-}$ no-Trp (5) and $Cse^{-/-}$ no-Trp (8); for epididymal adipose, $Cse^{+/-}$ no-Cys (3) and $Cse^{-/-}$ no-Cys (4).

methionine-free (no-Met) and no-Cys diet, but not on a no-Met or control diet, at both days 3 and 7 (Extended Data Fig. 5a–e). To test whether this elevation influenced weight loss, we fed *Gdf15*-KO or *Fgf21*-KO mice a no-Met and no-Cys diet. We observed reduced weight loss in *Gdf15*-KO mice on day 1, although the difference diminished by day 7 (Extended Data Fig. 5f,g). Similarly, *Fgf21*-KO mice also exhibited a significant attenuation of weight loss (Extended Data Fig. 5h,i), arguing that both hormones contribute to weight loss on a no-Cys diet.

The ISR-dependent gene *Trib3* was notably upregulated in the liver of cysteine-deprived animals. Acetyl-CoA carboxylase 1 (ACC1), the rate-limiting enzyme in FA biosynthesis, which is negatively regulated by TRIB3[46], was markedly reduced in $Cse^{-/-}$ mice compared with in $Cse^{+/-}$ mice on the no-Cys diet (Figs. 3b and 4f), further suggesting a reduction

in de novo lipogenesis in the liver. Accordingly, there was significant reduction in serum TGs and FFAs on day 2 in $Cse^{-/-}$ compared with $Cse^{+/-}$ mice on the no-Cys diet (Fig. 4g,h).

Given the significant changes in the liver after Cys restriction, we tested whether restoring *Cse* expression in a liver-specific manner can rescue the phenotype. Administration of adeno-associated virus AAV8 with *TBG-CSE* completely prevented weight loss and preserved liver GSH and serum TG and FFA levels (Extended Data Fig. 6). Redistribution of GSH produced in the liver may contribute to the mitigation of the effects of cysteine deprivation.

To confirm that weight loss in the setting of cysteine deficiency is due to the concurrent induction of the ISR and OSR, we fed mice a no-Trp diet while administering L-buthionine sulfoximine (BSO), a specific

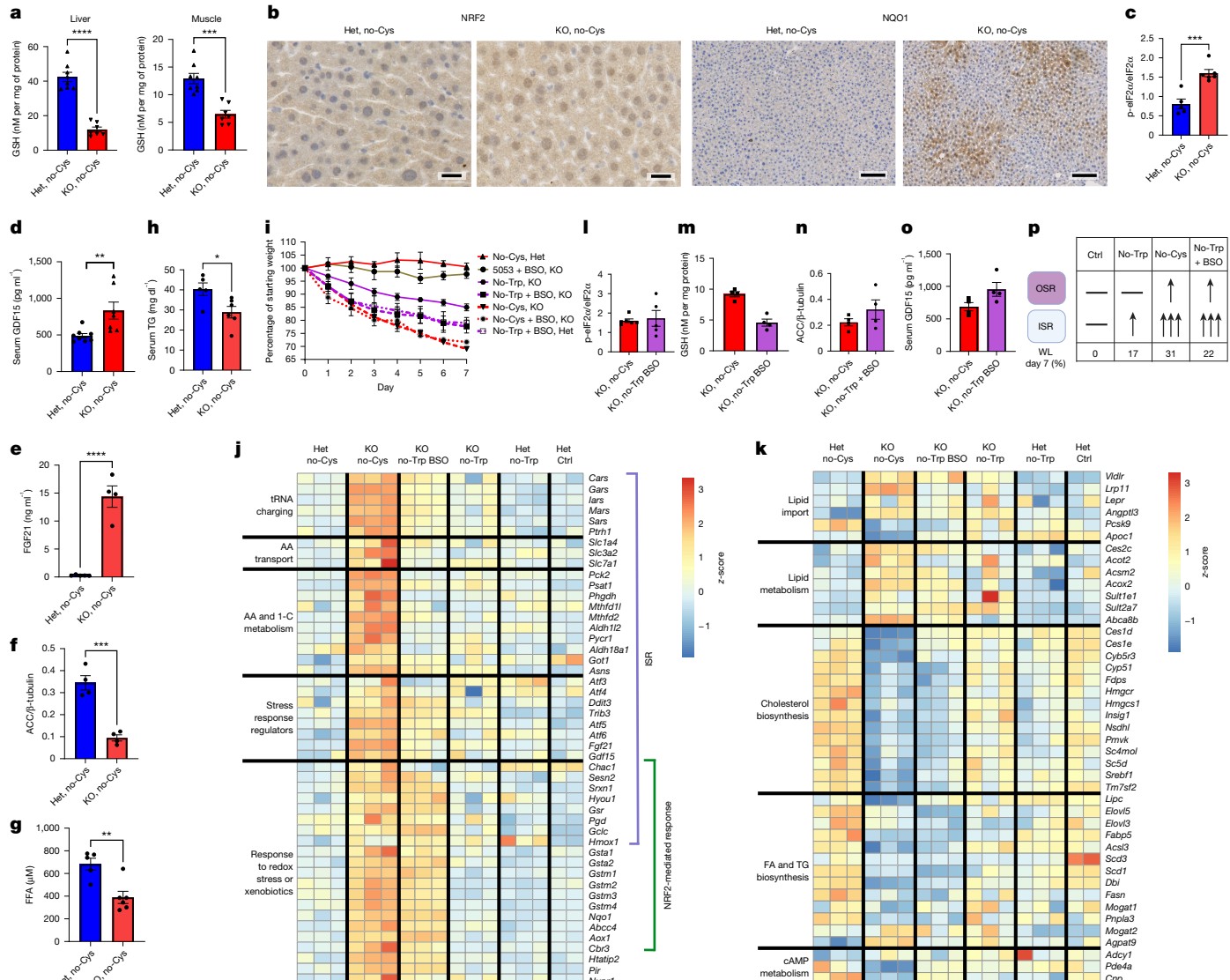

**Fig. 4 | General EAA deficiency coupled with a deficiency in GSH partially phenocopies cysteine deficiency. a**, GSH levels in liver and muscle of CR male *Cse*⁺/⁻ and *Cse*⁻/⁻ mice at day 2 on the no-Cys diet. *n* = 7 per group. **b**, Representative IHC staining (*n* = 4 per group) of NRF2 and NQO1 in the liver of *Cse*⁺/⁻ and *Cse*⁻/⁻ mice on a CR no-Cys diet at day 3. **c**, The p-eIF2α/eIF2α ratio in *Cse*⁺/⁻ and *Cse*⁻/⁻ mice on a CR no-Cys diet at day 2. *n* = 5 (*Cse*⁺/⁻) and *n* = 6 (*Cse*⁻/⁻). **d,e**, The GDF15 (**d**) and FGF21 (**e**) serum levels at day 2 of CR no-Cys diet. *n* = 8 (*Cse*⁺/⁻, GDF15), *n* = 7 (*Cse*⁻/⁻, GDF15), *n* = 5 (*Cse*⁺/⁻, FGF21) and *n* = 4 (*Cse*⁻/⁻, FGF21). **f**, ACC1 protein levels normalized to β-tubulin in *Cse*⁺/⁻ and *Cse*⁻/⁻ mice at day 2 on a CR no-Cys diet. *n* = 4 per group. **g,h**, The FFA (**g**) and TG (**h**) serum levels at day 2 of a CR no-Cys diet. *n* = 5 (*Cse*⁺/⁻), *n* = 6 (*Cse*⁻/⁻). **i**, Body-weight measurements of *Cse*⁺/⁻ and *Cse*⁻/⁻ mice on different diets. *n* ≥ 4 in each group.

**j,k**, Representative bulk liver mRNA-seq data for multiple groups represented as a heat map for genes in ISR and OSR (**j**), and those related to cholesterol and lipid metabolism (**k**). A complete list of differentially expressed genes for all sequenced samples is presented in Supplementary Table 2. **l–o**, The p-eIF2α/eIF2α ratio in the liver (**l**), liver GSH (**m**), ACC1 protein levels normalized to β-actin (**n**) and serum GDF15 (**o**) in CR *Cse*⁻/⁻ mice on a no-Cys diet compared with CR *Cse*⁻/⁻ mice on a no-Trp + BSO diet at day 2. *n* = 5 (**l**) and *n* = 4 (**m–o**) per group. **p**, Summary of the effects of no-Trp, no-Cys and no-Trp + BSO diets in *Cse*⁻/⁻ mice. WL, weight loss. The diagram in **p** was created using BioRender. Statistical analysis was performed using unpaired *t*-tests (**a** and **d–h**). Data are mean ± s.e.m. **\*\*P* < 0.01, \*\*\*\*P* < 0.0001. Scale bars, 20 μm (left) and 100 μm (right).

inhibitor of γ-glutamyl cysteine synthase, which decreases GSH but not cysteine (Fig. 1a). By day 3, the no-Cys and no-Trp + BSO diets resulted in approximately 20% and 18% weight loss, respectively, in *Cse*⁻/⁻ mice. However, by day 7, the no-Trp + BSO group had lost only 22%, in comparison to 31% in the no-Cys group, leaving a 9% unexplained difference in weight loss (Fig. 4i). We therefore compared the transcriptional response in livers of *Cse*⁻/⁻ mice on no-Cys versus no-Trp + BSO diets (Fig. 4j,k). The transcriptional upregulation of ISR was slightly weaker in the no-Trp + BSO group compared with the no-Cys group, notwithstanding the similar level of p-eIF2α (Fig. 4j,l).

Consistent with the reduction in liver GSH of *Cse*⁻/⁻ mice on no-Trp + BSO and no-Cys diets, there was highly similar upregulation of

OSR in both groups (Fig. 4j,m). Inclusion of BSO also substantially downregulated cholesterol and FA biosynthesis genes, aligning with previous studies involving GSH depletion[47] (Fig. 4k). By day 2, liver ACC1 levels, as well as the levels of serum GDF15, were comparable between *Cse*⁻/⁻ mice on either no-Trp + BSO or no-Cys diets (Fig. 4n,o). These results indicate that liver GSH primarily controls the OSR and de novo FA and cholesterol biosynthesis.

## Cys depletion leads to lower CoA

CoA has a central role in energy metabolism. CoA deficiency is expected to cause substantial abnormalities in cellular metabolism and can

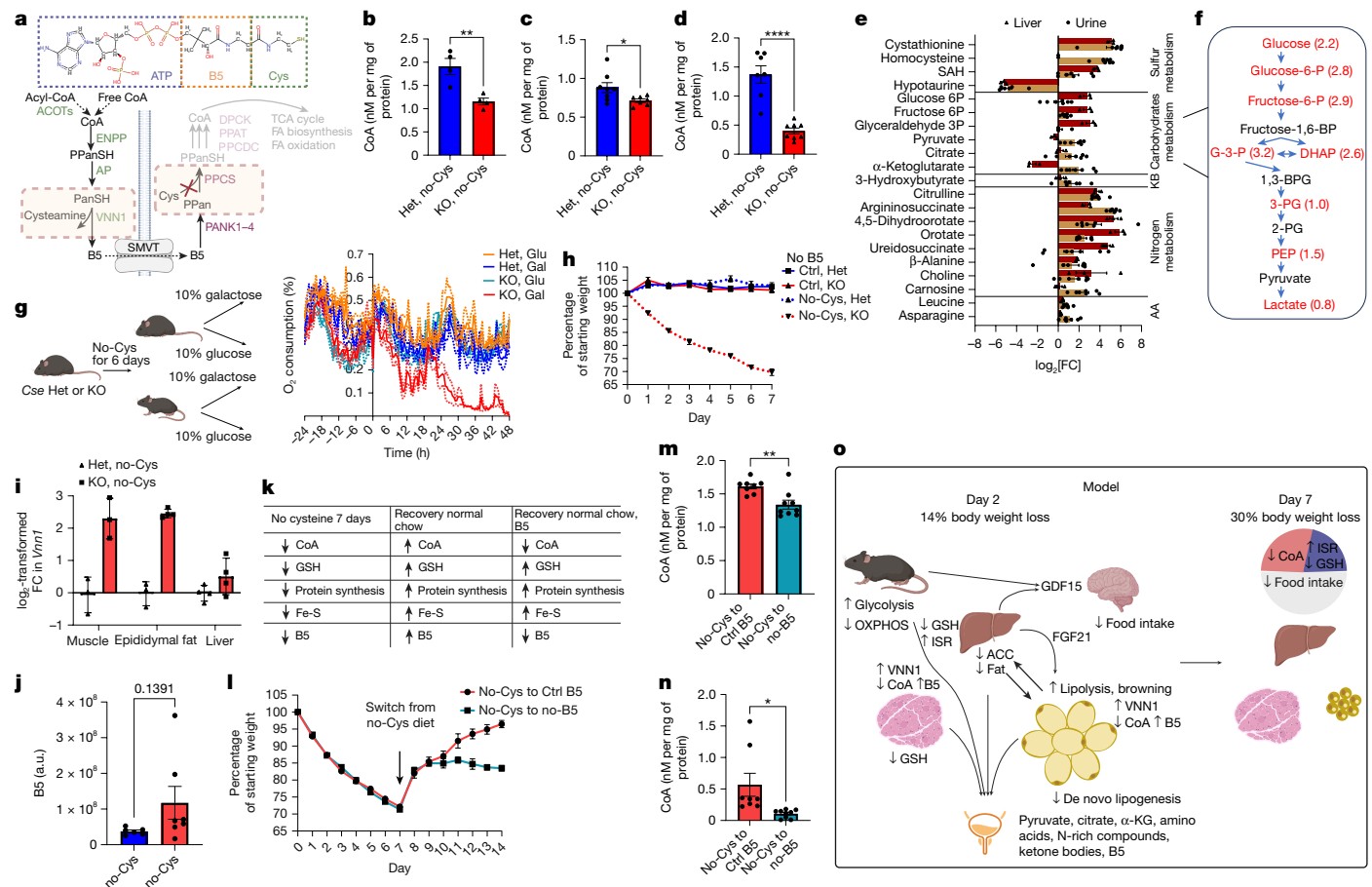

**Fig. 5 | Cysteine deficiency leads to metabolic inefficiency by depleting CoA.**
**a**, CoA structure and pathways for its production and degradation. PPanSH, phosphopantetheine; PanSH, pantetheine; PPan, phosphopantothenate; TCA, tricarboxylic acid. **b–d**, CoA levels in the liver (**b**) and muscle (**c**) at day 2 and in the liver at day 7 (**d**) from $Cse^{+/-}$ and $Cse^{-/-}$ mice on a no-Cys diet. $n \geq 4$ for all groups. **e,f**, Quantification (**e**) and glycolytic pathway schematic (**f**) of metabolite changes in the urine and liver of $Cse^{-/-}$ mice compared with $Cse^{+/-}$ mice on a no-Cys diet. $n = 6$ ($Cse^{+/-}$, urine), $n = 7$ ($Cse^{-/-}$, urine), $n = 4$ ($Cse^{+/-}$, liver) and $n = 3$ ($Cse^{-/-}$, liver). KB, ketone bodies. $P < 0.05$ was observed for all metabolites, except for 3-hydroxybutyrate ($P = 0.0891$) (**e**). For **f**, liver glycolytic intermediates that are upregulated in KO are shown in red, with $P < 0.05$ and $\log_2$[fold change] in brackets. **g**, $O_2$ consumption by mice that were provided with galactose (gal) or glucose (glu) as a sole carbon source after feeding on a

no-Cys diet. $n = 4$. **h**, Weight measurements of $Cse^{+/-}$ and $Cse^{-/-}$ mice on a no-B5 diet or a no-B5 and no-Cys diet. $n = 4$ per group. **i**, Changes in $Vnn1$ mRNA expression levels across tissues at day 2 of a CR no-Cys diet (Supplementary Tables 2–4). **j**, Urine B5 levels in $Cse^{+/-}$ and $Cse^{-/-}$ mice at day 2 of a CR no-Cys diet. a.u., arbitrary units. **k**, The expected levels of various cysteine-containing metabolites under different conditions. **l**, Weight measurements of $Cse^{-/-}$ mice after 7 days of a no-Cys diet followed by 7 days of either a control diet or a no-B5 diet. $n = 5$ per group. **m,n**, Liver (**m**) and muscle (**n**) CoA levels in mice on day 14 from the experiment shown in **l**. $n = 8$ ($Cse^{+/-}$) and $n = 9$ ($Cse^{-/-}$) in **m** and **n**. **o**, Model summary. The diagrams in **a**, **g**, **k** and **o** were created using BioRender. CoA structure in **a** obtained from https://www.rcsb.org/chemical-sketch (Marvin JS by Chemaxon). Statistical analysis was performed using unpaired $t$-tests (**b–f**, **j**, **m** and **n**). Data are mean ± s.e.m.

potentially explain the additional 9% weight loss in $Cse^{-/-}$ mice on the no-Cys diet compared to no-Trp + BSO diet (Fig. 4p). CoA is synthesized through the condensation of B5, cysteine and ATP and is thought to be highly stable[48] (Fig. 5a). While CoA levels are typically not impacted by B5 deficiency, the effect of cysteine restriction on CoA pools has not been studied.

Our experiments revealed lower total CoA levels in cysteine-deprived $Cse^{-/-}$ mice. By day 2, there was a notable 30% decrease in liver CoA levels and a weak but significant 15% decrease in muscle, compared with $Cse^{+/-}$ mice. This decline was even more pronounced by day 7, with a 75% reduction in liver CoA levels in $Cse^{-/-}$ mice on the no-Cys diet (Fig. 5b–d).

To further understand the metabolic changes linked to reduced CoA, we examined the liver and urine metabolomes of $Cse^{-/-}$ and $Cse^{+/-}$ mice after 2 days on a no-Cys diet. There were marked increases in the urine levels of pyruvate, citrate and α-ketoglutarate, metabolites that are involved in reactions preceding CoA-requiring steps in the tricarboxylic acid cycle (Fig. 5e, Extended Data Fig. 7a,b and Supplementary Table 5).

Furthermore, the livers of $Cse^{-/-}$ mice displayed marked accumulation of nearly all intermediates in the glycolysis pathway (Fig. 5f and Extended Data Fig. 7a).

The elevated levels in urine of multiple amino acids as well as intermediates in the urea cycle and pyrimidine biosynthesis indicate their diminished use for energy production (Fig. 5e and Extended Data Fig. 7a,c). However, the presence of ketone bodies in the urine suggests that, despite the low CoA, CoA-dependent FA degradation proceeds unimpeded (Fig. 5e). The increased liver levels of acyl-carnitines suggests futile attempts to recycle free CoA from acyl-CoA, and serve as alternative source of energy for brown fat[49] (Extended Data Fig. 7d).

Consistent with the reduced CoA, basal $O_2$ respiration was significantly decreased in lymph node T cells from $Cse^{-/-}$ mice after 7 days of cysteine deprivation, but not in those deprived of tryptophan. This suggests a compromised capacity for oxidative phosphorylation (Extended Data Fig. 7e,f). To further investigate the systemic effect of low CoA on mitochondrial activity, we restricted $Cse^{-/-}$ mice to either 10% glucose or

10% galactose solution as their sole energy source in the drinking water after 6 days on control, no-Cys or no-Trp diets (Fig. 5g). In contrast to glucose, galactose does not provide net ATP through glycolysis, forcing mice to rely entirely on oxidative phosphorylation for survival. Within 24 h of receiving only galactose, $Cse^{-/-}$ mice on the no-Cys diet showed a significant drop in $O_2$ consumption. These mice, in contrast to those on control or no-Trp diets, had to be euthanized within 42 h due to their inability to survive on galactose as their sole energy source (Fig. 5g and Extended Data Fig. 7g,h). As expected, no weight loss was observed with a B5-free diet, and there was no additional effect when B5 was removed from the cysteine-free diet. This underscores cysteine as the primary limiting factor for CoA biosynthesis (Fig. 5h).

To learn how carbohydrate metabolism changed with CoA depletion, we performed $^{13}$C-glucose tracing in $Cse^{+/-}$ and $Cse^{-/-}$ mice deprived of cysteine for 3 days (Extended Data Fig. 8a and Supplementary Table 6). Consistent with inefficient conversion of pyruvate to acetyl-CoA, there was a clear trend for elevated m+3 lactic acid in the liver of Cse$^{-/-}$ mice (Extended Data Fig. 8b,c). There was also a significant increase in urine of m+3 orotate, which is derived from pyruvate carboxylation to oxaloacetate, demonstrating another mechanism by which $Cse^{-/-}$ mice can waste carbon in the absence of CoA (Extended Data Fig. 8b,d).

Collectively, these findings suggest that, after cysteine deprivation, the carbon skeletons from dietary glucose and amino acids are lost in the urine in the form of intermediary metabolites, while lipid stores are used for ATP production.

There was also a significant increase in m+2 creatine and its precursor, guanidoacetic acid, in both the liver and urine of $Cse^{-/-}$ mice (Extended Data Fig. 8b,e–h). Given their substantial fat loss, elevated futile creatine cycling could contribute to thermogenesis during the observed weight loss[50]. Unexpectedly, we also detected upregulation of the cytoplasmic creatine kinase gene ($Ckm$) along with the suppression of its mitochondrial isoform ($Ckmt$), in the epididymal fat of $Cse^{-/-}$ mice (Extended Data Fig. 8i). Moreover, peak energy expenditure was significantly increased in $Cse^{-/-}$ mice on a no-Cys diet, alongside an increase in total liver creatine level by day 6 (Extended Data Fig. 8j,k). This suggests that the compromised mitochondrial activity due to CoA deficiency necessitates cytoplasmic futile creatine cycling for thermogenesis.

In mice deprived of cysteine or placed on a no-Trp diet + BSO, we also detected a slight but consistent decrease in the expression of multiple genes associated with mitochondrial respiratory complexes. This may be an adaptive mechanism to mitigate reactive oxygen species (Extended Data Fig. 9).

There are multiple ways for Cys deficiency to induce metabolic inefficiencies. Besides depleting CoA, low Cys availability can compromise Fe-S cluster assembly, downregulate mitochondrial protein biosynthesis or disturb the redox balance through low GSH. The upregulation of a pantetheinase $Vnn1$ in adipose and muscle tissues of $Cse^{-/-}$ mice on the no-Cys diet suggests increased degradation of pantetheine into B5 and cysteamine[48] (Fig. 5a,i). Normally, $Vnn1$ is activated under starvation and fasting conditions to redirect CoA from peripheral tissues to the liver[48]. However, in the context of cysteine deficiency, the inability to resynthesize CoA in $Cse^{-/-}$ mice results in CoA insufficiency and urinary loss of B5 (Fig. 5j), implying that, by day 7 on a no-Cys diet, along with the depletion of other cysteine containing molecules, the $Cse^{-/-}$ mice would also have lower B5 (Fig. 5k).

Indeed, when we reverted cysteine-deprived $Cse^{-/-}$ mice to a cysteine-sufficient but B5-deficient diet, they failed to gain as much weight as mice on the control B5 and cysteine-sufficient diet (Fig. 5l). Accordingly, GSH was fully rescued in the liver and partly in the muscle, while CoA levels were still significantly lower on the B5-deficient diet (Fig. 5m,n and Extended Data Fig. 10a,b). Furthermore, restoring B5 in drinking water after 7 days on the B5-deficient diet promptly rescued the difference in weight recovery (Extended Data Fig. 10c). Taken together, our results indicate that the loss of CoA contributes to metabolic inefficiency and weight loss on a no-Cys diet (Fig. 5k). The lack of CoA alone probably leads to significant changes in metabolism, contributing to rapid weight loss and preventing weight gain, showing that CoA is a major regulator of metabolic efficiency.

## Discussion

Our results indicate that cysteine deprivation triggers a global reprogramming of metabolic processes, culminating in rapid and readily reversible weight loss through the decline in adipose tissue lipid content, reduced lipogenesis and excretion of intermediary metabolites that cannot be effectively used (Fig. 5o). These results extend earlier studies that demonstrated an essential role of cysteine in $Cse^{-/-}$ mice and also reported weight loss without compromising liver function, albeit at a much slower rate with a low-cysteine diet[12,13]. Our results show that cysteine, rather than methionine, mediates the benefits of SAAR. Another group has also demonstrated similar findings using a different Cse KO background and a different cysteine-free diet[51], highlighting the high reproducibility of these results.

Owing to its potent cytotoxicity[52], the cellular concentration of cysteine is the lowest of all amino acids[53]. In mammals, most dietary cysteine is absorbed by the liver and is rapidly converted into much safer molecules—GSH and taurine[54]. When animals are exposed to a no-Cys diet, intracellular cysteine is rapidly depleted due to protein, GSH and CoA synthesis. Thus, a combination of low cysteine concentration, a high demand across multiple cellular processes and oxidative stress mediated by a reduction in GSH probably explains the amplified transcriptional response to cysteine deficiency compared with other amino acid deficiencies. This may also clarify why, despite similar p-eIF2α levels, the no-Trp diet combined with BSO did not elicit an identical transcriptional response in ISR-associated genes.

Our findings suggest that millimolar GSH levels in tissues do not prevent weight loss by acting as a cysteine reservoir. Consequently, animals reduce metabolism by lowering CoA levels and suppressing protein synthesis, avoiding the rapid release of cytotoxic cysteine. This aligns with human studies linking high cysteine levels to obesity, neurological disorders and cardiovascular diseases[55–59]. Supporting evidence from rodents and nematodes shows that restricting sulfur amino acids can extend lifespan[23,25,26].

The upregulation of vanin ($Vnn1$), results in rapid depletion of CoA through its degradation into cysteamine and B5, which cannot be effectively reused in the liver due to the lack of cysteine. The use of FAs rather than carbohydrates when CoA levels are limited could potentially be driven by a lower $K_m$ for acyl-CoA synthetases compared with that for pyruvate dehydrogenase[60,61]. Thus, as CoA becomes limiting, lipids could become the preferred energy source leading to the lower RER observed. Our study demonstrates that cysteine-deficiency-induced CoA loss results in metabolic reprogramming and inefficiency, leading to urinary excretion of both glycolytic and tricarboxylic-acid-cycle intermediates, ketone bodies and other biosynthetic intermediates, contributing to the observed weight loss.

Both GSH depletion and ISR also suppress mitochondrial activity, reducing reactive oxygen species production and preventing the formation of new mitochondria[62,63]. Thus, the combined effect of ISR, GSH loss and CoA deficiency probably contributes to the overall mitochondrial inefficiency.

In summary, our findings unravel the profound impact of rendering a non-EAA essential and subsequently removing it from the diet, leading to swift and substantial fat loss. These observations have important implications for the field of metabolic medicine, especially in the context of obesity management.

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

# Methods

## Mice

Mice were bred and maintained in the Alexandria Center for the Life Sciences animal facility of the New York University School of Medicine, under specific-pathogen-free conditions and were fed LabDiet standard 5053 diet before experimentation. C57BL/6 (Jax, 000664), *Gcn2*-KO (*B6.129S6-Eif2ak4tm1.2Dron/J*) and *Fgf21*-KO (*B6.129Sv(Cg)-Fgf21tm1.1Djm/J*) mice were purchased from Jackson Laboratories. *Cse*-KO (129/C57BL/6 background) mice were generated by R. Wang as previously described and provided by C. Hine[13]. Mice in all the experiments were at least 9 weeks old at the starting point of various diets unless described specifically. Heterozygous and KO mice were co-housed except during food intake measurements and calorie restriction experiments. We have included a summary of power analysis for main figures and extended data figures as Supplementary Table 7. All animal procedures were performed in accordance with protocols approved by the Institutional Animal Care and Usage Committee of New York University School of Medicine. Tissues and samples were harvested and analysed in a blinded manner.

## Generation of *Gdf15*-KO mice

*Gdf15*-KO mice (C57BL/6 background) were generated by the Genome Editing Shared Resource of Rutgers Cancer Institute. Deletion of *Gdf15* was performed by co-injecting two CRISPR gRNAs flanking the exon 2 region with Cas9 protein into C57BL/6J zygotes. Verification of CRISPR knockout for the 1.3 kb deletion was performed by PCR. In brief, the primers GDF15A 5′-TCAACTTTAAGCCAGAAGGTGGCG-3′, GDF15B 5′-CTTCGGGGAGACCCTGACTCAGC-3′ and GDF15D 5′-ACTGC GAATCTAGAGAACCCTGAC-3′ were used to amplify the targeted region. Tail snips from founder mice were submitted for Sanger sequencing to confirm homozygous deletion. Founder mouse had a deletion of 1,304 bp, which encompasses all of exon 2, the pro-peptide region of GDF15.

## Diets

All custom diets were procured from TestDiet. All diets were based on the defined amino acid diet 5CC7. For individual or dual amino-acid-depleted diets, the specific amino acids were completely removed, and all other amino acids were increased in proportion. For the B5-deficient diet, it was removed from the same 5CC7 defined diet. Unless specified control diet refers to the 5CC7 diet. D-Galactose (Sigma-Aldrich, G0750) and D-glucose (Sigma-Aldrich, G8270) were purchased from Sigma-Aldrich and were dissolved in water and filter-sterilized. A list of diet names is available on request. All weight-loss curves are plotted with respect to starting body weight and are represented as the mean ± s.d.

## Food-intake measurements

Initially, food-intake measurements were conducted using metabolic cages; however, mice repeatedly left food debris on the cage bottom compromising the consumption measurements. To address this issue, individually housed mice were provided with a restricted amount of pellets (15–20 g), and the remaining food was measured every 24 h. Additional intact pellets were added as needed. Mice that left debris on the cage bottom on any day were excluded from the analysis.

## Histology and immunohistochemistry

Mice were first anaesthetized with ketamine and xylazine. They were then perfused, first with chilled PBS, followed by 4% PFA in chilled PBS. Tissues were then collected and fixed for 24 h in 4% PFA in PBS at room temperature and then transferred to 70% ethanol. Tissues were then paraffin embedded. Sections (5 μM thick) were cut and stained with H&E. Tissues for immunostaining underwent deparaffinization followed by antigen retrieval for 20 min at 100 °C with Leica Biosystems ER2 solution (pH9, AR9640) and endogenous peroxidase activity blocking with $H_2O_2$. Sections were incubated with primary antibodies against UCP1 (CST, 72298S, E9Z2V, 1:1,000), CASP3 (CST, 9579S, D3E9, 1:200), NQO1 (Sigma-Aldrich, HPA007308, 1:100), custom NRF2 (1:1,000, provided by E. Schmidt)[64]. Primary antibodies were detected with anti-rabbit HRP-conjugated polymer and 3,3′-diaminobenzidine substrate, followed by counterstaining with haematoxylin, all of which were provided in the Leica BOND Polymer Refine Detection System (DS9800).

## Seahorse assay

Cells were isolated from inguinal, brachial and mesenteric lymph nodes. After ACK lysis to remove RBCs, the CD8+ T cells, B cells and CD11c+ cells were bead-depleted to leave primarily CD4+ T cells. All steps were done at 4 °C and used dialysed fetal bovine serum to avoid amino acids in the serum. Seahorse 24-well plates were plated with Cell-Tak (Corning) according to the manufacturer's instructions and 100,000 T cells were plated in 500 μl of room-temperature Seahorse medium. After 30 min at 37 °C, cells were analysed using the Seahorse XFe24 Analyzer.

## Indirect calorimetry

To assess possible changes in metabolic parameters, mice were measured by indirect calorimetry using an open respirometry system (TSE PhenoMaster, TSE Systems). Mice were weighed and individually housed in the specialized home cages within a temperature- and humidity-controlled climate chamber (22 ± 0.5 °C, 50 ± 1% relative humidity). The light cycle was set to 12:12 (lights on at 06:30) and the air flow rate for each cage was set to 0.35 l min$^{-1}$ (with 0.25 l min$^{-1}$ diverted to the gas sensors during the sampling period: 190 s line purge and 10 s active sample). Oxygen consumption (vO$_2$, ml h$^{-1}$), carbon dioxide production (vCO$_2$, ml h$^{-1}$) and water intake (ml), and activity (beam break counts, $x + y + z$ axes) were recorded at 30 min intervals. The RER (RER = vCO$_2$/vO$_2$) was calculated from measured vO$_2$ and vCO$_2$ values. Mice were first acclimatized to the metabolic home cages for 24 h before data were collected for analysis. Mice were housed until the set end point of the experiment. Data were acquired and exported with TSE PhenoMaster software v.8.1.4.14156 (TSE Systems; https://www.tse-systems.com/products/phenomaster/). After data export, data were uploaded to CalR (https://CalRapp.org/)[65] for visualization and analysis.

## Preparation of tissue lysates and metabolite analysis

Flash-frozen mouse tissues were homogenized in PBS supplemented with protease (cOmplete, Roche) and phosphatase (PhosSTOP, Roche) inhibitors using the IKA Ultra Truemax T8 homogenizer on ice. Aliquots of homogenates were used for RNA isolation immediately. Cells in homogenized tissues were lysed by two freeze–thaw cycles and subsequent sonication (Diagenode Bioruptor). The protein concentration in the lysates was determined by BCA assay. Aliquots of lysates were filtered through 10 kDa Amicon filters, and glutathione and CoA were measured in the flow-through using the following kits: Glutathione detection kit (Cayman, 703002) and CoA detection kit (Abcam, ab102504). Metabolite concentration was normalized to the protein concentration in lysates. GDF15 and FGF21 were measured in the mouse serum by ELISA kits (Abcam, ab216947; and R&D Systems, MF2100). TG and FAA in the mouse serum were analysed using the Cayman (10010303) and Sigma-Aldrich (MAK466) kits, respectively.

## Sequencing and differential expression analyses

To study transcriptional response in mouse tissues, total RNA was isolated from the homogenates using the trizol (1:10 ratio) method. mRNA was purified by NEBNext Poly(A) mRNA Magnetic Isolation Module (NEB, E7490S) from Turbo-DNase-treated total RNA. A NEB Next Ultra Library Preparation Kit (NEB, E7530S) was used to prepare 0.5 μg of total RNA for RNA-seq. At least three to five animals were used for each experimental condition. The libraries were sequenced using the Illumina NextSeq 200 instrument in a paired-end 2 × 61 cycles set-up to a

depth of 20–50 million reads per sample. The reads were aligned against mouse genome assembly GRCm38 using Hisat2 v.2.1.0. The number of reads in annotated genes was counted using htseq-count version .11.069. The differential gene expression analysis and visualization were performed with DESeq2 version 1.48 using Wald test[66,67]. Heat maps were generated on the basis of $z$-scores of normalized counts using the R library pheatmap. GO analysis is available at the Gene Ontology web application (http://geneontology.org/).

## Western blotting
Tissue lysates supplemented with LDS loading buffer and 10 mM DTT were heated for 5 min at 95 °C. Proteins were separated on Bis-Tris SDS–PAGE, transferred onto nitrocellulose membrane and probed with anti-p-eIF2α (Cell Signaling Technology, 3597), anti-eIF2α (Cell Signaling Technology, 2103) anti-p-ACC (Cell Signaling Technology, 3661), anti-ACC (Cell Signaling Technology, 3662), anti-β-tubulin (Proteintech, 10068-1-AP) and anti-β-Actin–Peroxidase antibody (Sigma-Aldrich, A3854).

## MS analysis
**Sample preparation.** *Stool samples.* Stool samples and food pellets were weighed into bead-blaster tubes containing zircon beads. Extraction buffer containing 80% methanol with 500 nM metabolomics amino acid standard mix (Cambridge Isotopes Laboratory) was added to each to reach a final concentration of 10 mg ml$^{-1}$. The samples were homogenized using D2400 BeadBlaster homogenizer (Benchmark Scientific) then centrifuged at 21,000$g$ for 3 min. Then, 450 µl of metabolite extract was transferred to a new 1.5 ml Eppendorf tube and dried using the Speedvac. The samples were reconstituted in 50 µl of mass spectrometry (MS)-grade water and sonicated for 2 min. The samples were then centrifuged at 21,000$g$ for 3 min. The samples were transferred to glass LC vials for analysis by LC–MS.
*Liver samples.* Approximately 300 mg of liver was homogenized in 1 ml of PBS and then subjected to three freeze–thaw cycles. Liver samples were filtered using a 10 kDa filter. Protein concentrations before deproteinization were measured using BSA standard curve and found to be between 12.8–42.0 mg ml$^{-1}$. For metabolomics extracts, on average protein concentration was determined to be 28.58 mg ml$^{-1}$ per 300 mg of tissue. This value was used to scale all liver extracts to each other. Scaled liver extracts were transferred to bead blaster tubes with zircon beads and homogenized using D2400 BeadBlaster homogenizer (Benchmark Scientific) in cold 80% methanol spiked with 500 nM metabolomics amino acid standard mix (Cambridge Isotopes Laboratory). The samples were centrifuged at 21,000$g$ for 3 min to pellet any insoluble materials. Then, 450 µl of metabolite extract was transferred to a new 1.5 ml Eppendorf tube and dried using the Speedvac. The samples were reconstituted in 50 µl of MS-grade water and sonicated for 2 min. The samples were then centrifuged at 21,000$g$ for 3 min. The samples were transferred to glass liquid chromatography (LC) vials for analysis by LC–MS.
*Urine samples.* Urine samples were collected and stored as frozen aliquots between around 5–7 µl. For metabolite extraction, 5 µl of urine was transferred to a glass insert and extracted using 195 µl of cold 80% methanol spiked with 500 nM metabolomics amino acid standard mix (Cambridge Isotopes Laboratory). Glass inserts were transferred into 1.5 ml Eppendorf tubes and centrifuged at 3,000$g$ for 10 min to pellet insoluble material. Then, 180 µl of extract was transferred to a 1.5 ml Eppendorf tube and dried completely using the Speedvac. To each tube, 20 µl of MS-grade water was added to reconstitute metabolites. The samples were sonicated for 2 min and then centrifuged for 3 min at 21,000$g$. Then, 15 µl of samples was transferred to a glass LC–MS vial for analysis.

**LC–MS/MS analysis.** The samples were subjected to LC–MS analysis to detect and quantify known peaks. A metabolite extraction was carried out on each sample using a previously described method[68].

The LC column was a Millipore ZIC-pHILIC (2.1 × 150 mm, 5 µm) coupled to a Dionex Ultimate 3000 system and the column oven temperature was set to 25 °C for the gradient elution. A flow rate of 100 µl min$^{-1}$ was used with the following buffers: (A) 10 mM ammonium carbonate in water, pH 9.0; and (B) neat acetonitrile. The gradient profile was as follows; 80–20% B (0–30 min), 20–80% B (30–31 min), 80–80% B (31–42 min). The injection volume was set to 2 µl for all analyses (42 min total run time per injection).

MS analyses were carried out by coupling the LC system to a Thermo Q Exactive HF mass spectrometer operating in heated electrospray ionization mode. The method duration was 30 min with a polarity-switching data-dependent top 5 method for both positive and negative modes. The spray voltage for both positive and negative modes was 3.5 kV and the capillary temperature was set to 320 °C with a sheath gas rate of 35, aux gas of 10 and maximum spray current of 100 µA. The full MS scan for both polarities used 120,000 resolution with an AGC target of $3 \times 10^6$ and a maximum IT of 100 ms, and the scan range was from 67–1,000 $m/z$. Tandem MS spectra for both positive and negative mode used a resolution of 15,000, AGC target of $1 \times 10^5$, maximum IT of 50 ms, isolation window of 0.4 $m/z$, isolation offset of 0.1 $m/z$, fixed first mass of 50 $m/z$ and three-way multiplexed normalized collision energies of 10, 35 and 80. The minimum AGC target was $1 \times 10^4$ with an intensity threshold of $2 \times 10^5$. All data were acquired in profile mode.

**Hybrid metabolomics data processing.** The resulting Thermo RAW files were converted to mzXML format using ReAdW.exe v.4.3.1 to enable peak detection and quantification. The centroided data were searched using a custom Python script Mighty_skeleton v.0.0.2 and peak heights were extracted from the mzXML files based on a previously established library of metabolite retention times and accurate masses adapted from the Whitehead Institute[69] and verified with authentic standards and/or high resolution MS/MS spectral manually curated against the NIST14MS/MS[70] and METLIN (2017)[71] tandem mass spectral libraries. Metabolite peaks were extracted based on the theoretical $m/z$ of the expected ion type, for example, [M+H]$^+$, with a ±5 parts per million tolerance, and a ±7.5 s peak apex retention time tolerance within an initial retention time search window of ±0.5 min across the study samples. The resulting data matrix of metabolite intensities for all samples and blank controls was processed using a custom statistical pipeline Metabolize v.1.0 and final peak detection was calculated based on a signal to noise ratio (S/N) of 3× compared with blank controls, with a floor of 10,000 (arbitrary units). For samples for which the peak intensity was lower than the blank threshold, metabolites were annotated as not detected, and the threshold value was imputed for any statistical comparisons to enable an estimate of the fold change as applicable. The resulting blank corrected data matrix was then used for all group-wise comparisons, and $t$-tests were performed using the Python SciPy (v.1.1.0)[72] library to test for differences and to generate statistics for downstream analyses. Any metabolite with $P < 0.05$ was considered to be significantly regulated (up or down). Heat maps were generated with hierarchical clustering performed on the imputed matrix values using the R library pheatmap (v.1.0.12; https://CRAN.R-project.org/package=pheatmap). Volcano plots were generated using the R library Manhattanly (v.0.2.0). To adjust for significant covariate effects (as applicable) in the experimental design the R package, DESeq2 (v.1.24.0)[67] was used to test for significant differences. Data processing for this correction required the blank corrected matrix to be imputed with zeroes for non-detected values instead of the blank threshold to avoid false positives. This corrected matrix was then analysed using DESeq2 to calculate the adjusted $P$ value in the covariate model.

## AAV-mediated rescue experiments
Mice were injected retro-orbitally with $2 \times 10^{11}$ liver-specific AAV8 TBG-CSE or AAV8 TBG-eGFP viral particles per mouse (Vector Biolabs). Then, 2 weeks after AAV administration, the mice were placed

onto a no-Cys diet and, after 1 week, tissues and serum were collected and analysed.

## Heavy glucose tracing

Mice were fasted for 18 h after 3 days of no-Cys diet. They were then orally given 1.25 mg per kg of D-Glucose-$^{13}C_6$ (Sigma-Aldrich) dissolved in water. Mice were euthanized at 45 min to collect livers or at 2 h to collect urine. Liver was processed as described in the 'Preparation of tissue lysates and metabolite analysis' section. Subsequently liver lysates and urine were analysed as described in the 'MS analysis' section.

The data were then processed, and naturally expected levels of each molecule's isotope were subtracted by the expected natural frequency of each isotope multiplied by each individual sample's lowest weight isotope amount for the molecule.

These corrected data were then used for the analysis and are included in Supplementary Table 6.

## Reporting summary

Further information on research design is available in the Nature Portfolio Reporting Summary linked to this article.

## Data availability

Sequencing data generated and used for this project are available at Gene Expression Omnibus under accession code GSE280181. Metabolomics data generated and used for this project are available at Metabolomics Workbench (project ID: PR002200; https://doi.org/10.21228/M87Z47). Raw gel images are provided in Supplementary Fig. 1. Source data are provided with this paper.

## Code availability

All analyses were performed with routinely used and previously described codes. No custom codes were used in this study. Codes that were used are available on request.

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

**Acknowledgements** We thank the members of the Nudler and Littman laboratories for their discussions and review of the manuscript; K. Laborc for assistance with the metabolic cage experiments; E. Schmidt for the custom NRF2 antibody; S. Y. Kim and the members of the NYU Rodent Genetic Engineering Laboratory (RGEL) for rederivation of mutant mice; C. Loomis and the staff at the Experimental Pathology Research Laboratory of NYULMC for histology and Immunohistochemistry and P. Cohen for discussions; and S. Gottesman for editing the manuscript. The Experimental Pathology Research Laboratory is supported by National Institutes of Health Shared Instrumentation grants S10OD010584-01A1 and S10OD018338-01. This work was supported by the NSERC RGPIN-2023-05099 (R.W.); Ludwig Princeton Branch, and Cancer Grand Challenges (NCI 1OT2CA278609-01, CRUK (CGCATF-2021/100022) (E.W.); ACS RSG-21-115-01-MM) and NIH R35GM147119 (M.E.P.); 24/7 metabolic home cage experiments were performed in the NYULMC Rodent Behavior Laboratory supported by the NIH BRAIN Initiative U19NS1076 (A.C.M.); Blavatnik Family Foundation (E.N.); Howard Hughes Medical Institute (E.N. and D.R.L.); and NIH R01AI158687 (D.R.L.). BioRender was used to make Figs. 2a, 3a, 4p and 5a,k and o and Extended Data Figs. 6a and 8a. CoA structure in Fig. 5a was obtained from https://www.rcsb.org/chemical-sketch (Marvin JS by Chemaxon).

**Author contributions** A.V., I.G., E.N. and D.R.L. conceived the project. A.V. and I.G. designed experiments and analysed data. A.V. performed mouse experiments and collected samples with help from D.D., Y.M. and M.P.; I.G. processed samples and performed biochemical analysis with help from A.V.; B.G.-L. and A.C.M. performed metabolic cage experiments. I.S. performed RNA-seq and analysis. R.J. and D.R.J. performed MS experiments and helped with analysis. M.G.-J., E.W., R.W. and T.P. provided resources. T.P. and M.E.P. advised on experiments. A.V., I.G., D.R.L. and E.N. wrote the manuscript with input from all of the authors. D.R.L. and E.N. acquired funding and supervised the research.

**Competing interests** D.R.L. consults for and has equity interest in Vedanta Bioscience, Sonoma Immunotherapeutics, Immunai, IMIDomics and Pfizer.

**Additional information**
**Correspondence and requests for materials** should be addressed to Dan R. Littman or Evgeny Nudler.

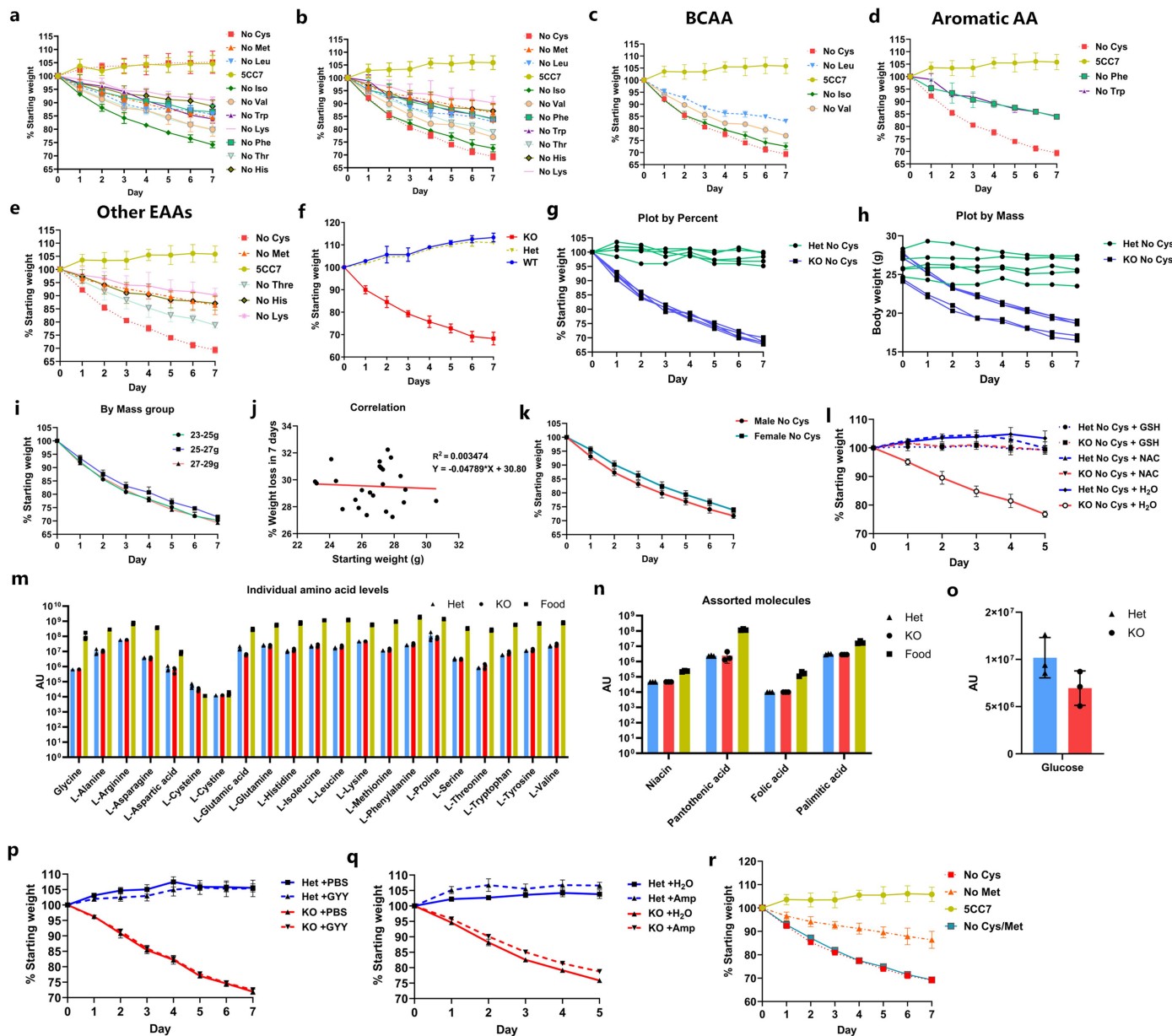

**Extended Data Fig. 1 | Characterization of weight loss induced by a cysteine-deficient diet and comparative analysis with other EAA deficiencies.** **a**, Het and **b**, KO weight curves with various essential amino acid deficiencies (n ≥ 4 per group). **c-e**, KO on various essential amino acid deficiencies by class: **c**, Branched chain amino acids; **d**, Aromatic amino acids; **e**, Other EAAs. **f**, Weight curves of WT compared to *Cse* Het and KO mice on No Cys diet. Weight loss curves of *Cse* KO on No Cys plotted individually by **g**, percentage **h**, mass. **i**, Weight loss curves by percentage plotted according to different starting weights (n ≥ 5 in all groups). **j**, Percentage weight loss at Day 7 of *Cse* KO mice on a No Cys diet, plotted against starting mass. **k**, Weight loss curves of male vs female *Cse* KO mice on a No Cys diet (n = 9 per group). **l**, Weight curves of female

Het and KO mice on a No Cys diet with either 0.4% NAC or 1% GSH in drinking water (n ≥ 4 per group). **m-o**, Levels of various nutrients in the administered diet (yellow) and in Het (blue) or KO (red) stool from microbiota-depleted mice fed on a No Cys diet for 3 days. **m**, Amino acid levels. **n**, Vitamins and palmitic acid levels. **o**, Glucose levels. **p**, Weight curves of female mice on a No Cys diet with or without daily injections of 40 mg/kg of GYY4137 (n ≥ 4 per group). **q**, Weight curves of female mice on a No Cys diet with and without antibiotic treatment (1 g/L Ampicillin) (n ≥ 4 per group). **r**, Weight curves of *Cse* KO mice on a No Met No Cys compared to No Cys, No Met, or control diets (n = 4 per group). Hets and KOs mice were co-housed for each experimental condition. Data are mean ± s.d.

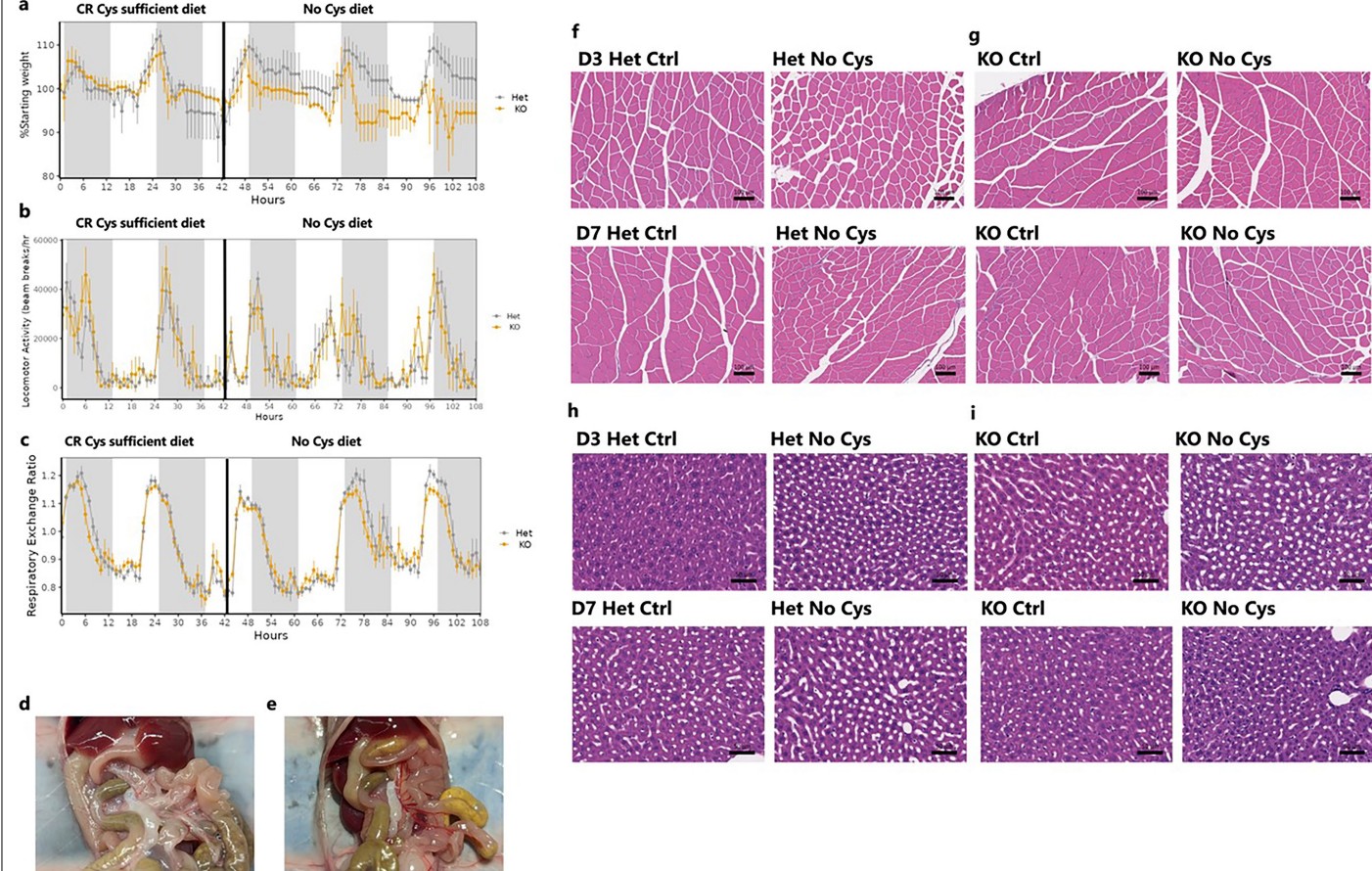

**Extended Data Fig. 2 | Metabolic profile in female mice and Gross histology of skeletal muscle and liver. a-c** Metabolic cage profiles of male *Cse* Het and KO mice on CR (n = 4 for Het and 4 for KO) with experimental design as in Fig. 2a, body weight measurements **a**, locomotor activity **b**, and respiratory exchange ratio **c. d, e**, Representative visceral fat image of Het (d) and KO (e) mice on No Cys diet after 3 days (n = 4 per group). **f-i**, Representative H&E staining of skeletal muscle (quadriceps) (f,g) and liver (h,i) from Het (f,h) and KO (g,i) mice on Days 3 and 7 of calorically restricted Ctrl or No Cys diets (n = 4 per group).

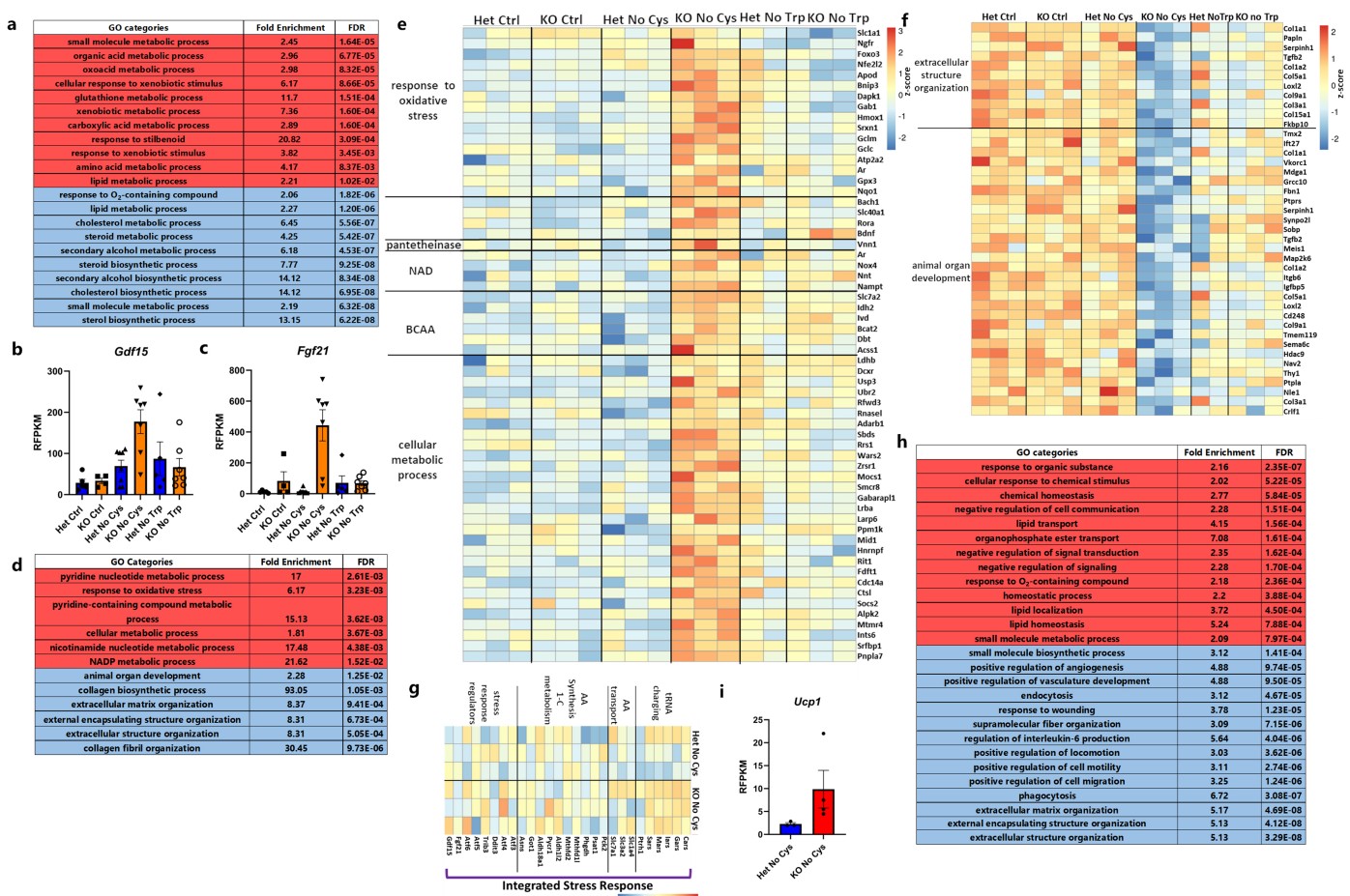

**Extended Data Fig. 3 | Transcriptional response to cysteine deficiency.**
**a**, GO categories that are significantly enriched among genes up- or down-regulated in liver of KO compared to Het mice on No Cys diet. **b,c**, RFPKM values for *Gdf15* (**b**) and *Fgf21* (**c**) genes from the livers of mice on the different diets (n ≥ 4 for all groups, See also Supplementary Table 2). **d**, GO categories that are significantly enriched among genes up- or down-regulated in muscle of KO compared to Het mice on No Cys diet. **e-g**, Muscle bulk RNA-Seq data presented as a heatmap focusing on: upregulated genes involved in oxidative stress, cellular metabolic process, branched chain amino acid metabolism, NAD, and pantetheinase (e), downregulated genes in extracellular organization and animal organ development (f), and genes in the ISR that appear minimally affected (g) in KO compared to Het mice at Day 2 on No Cys diet (See also Supplementary Table 3). **h**, GO categories that are significantly enriched among genes up- or down-regulated in epididymal fat pad of KO compared to Het mice on No Cys diet. (See also Supplementary Table 4). **i**, RFPKM values for *Ucp1* gene from epididymal fat pad (n = 3 for Het, n = 4 for KO). All data are on Day 2 based on mice shown in the RNA-seq schematic in Fig. 3a. Red and blue indicates GO categories among up- and down-regulated genes, respectively. Data are mean ± s.e.m.

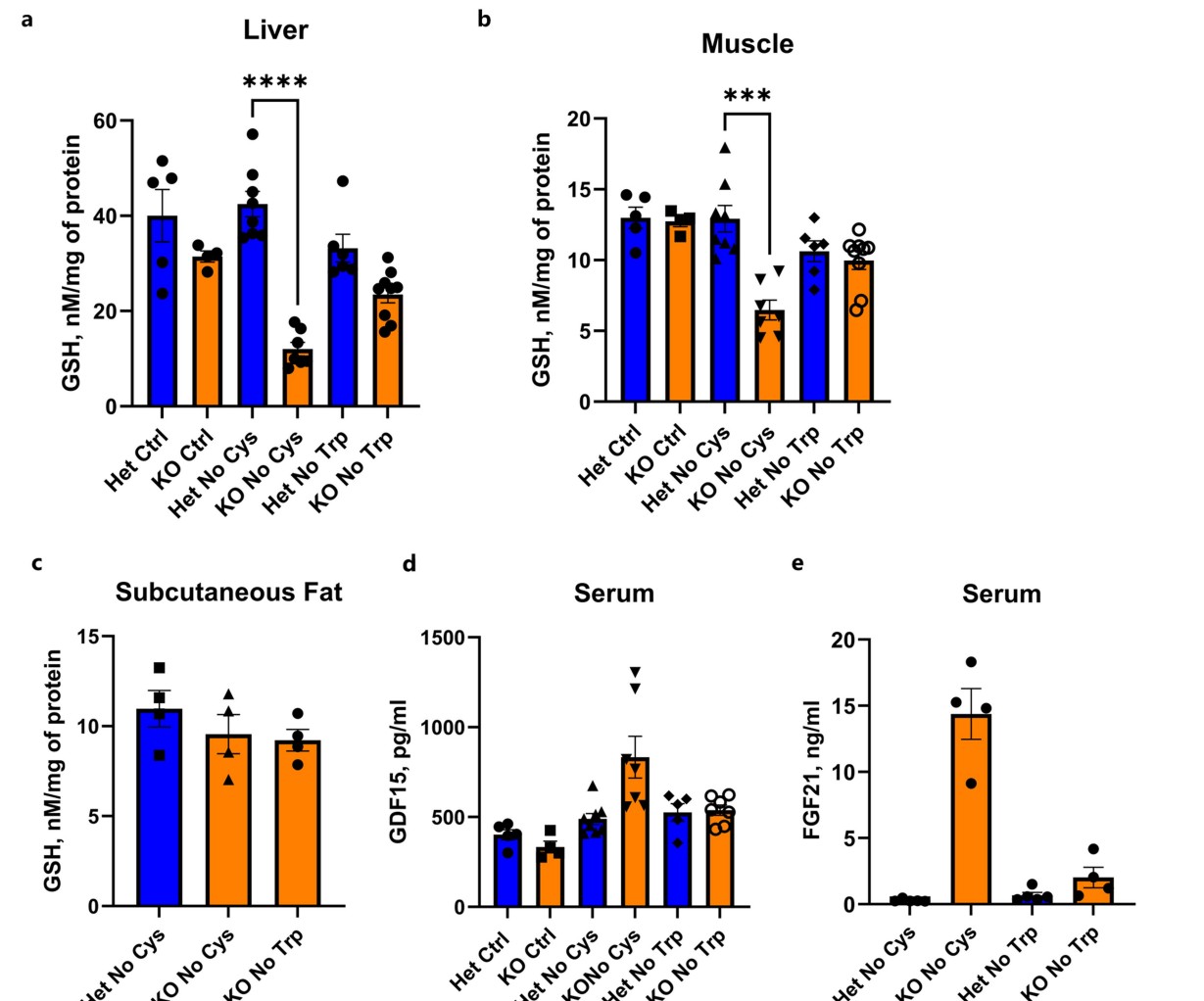

**Extended Data Fig. 4 | Effect of cysteine restriction on tissue GSH, serum GDF15 and FGF21 levels. a-e**, GSH levels in liver (a), muscle (b) and subcutaneous fat pad (c). GDF15 (d) and FGF21 (e) levels in serum from *Cse* Het and KO mice on Day 2 of CR diets without Cys or Trp or Ctrl diet (n ≥ 3 in each) Unpaired t-test, **a, b**, Data are mean ± s.e.m. **P < 0.01 and ****P < 0.0001.

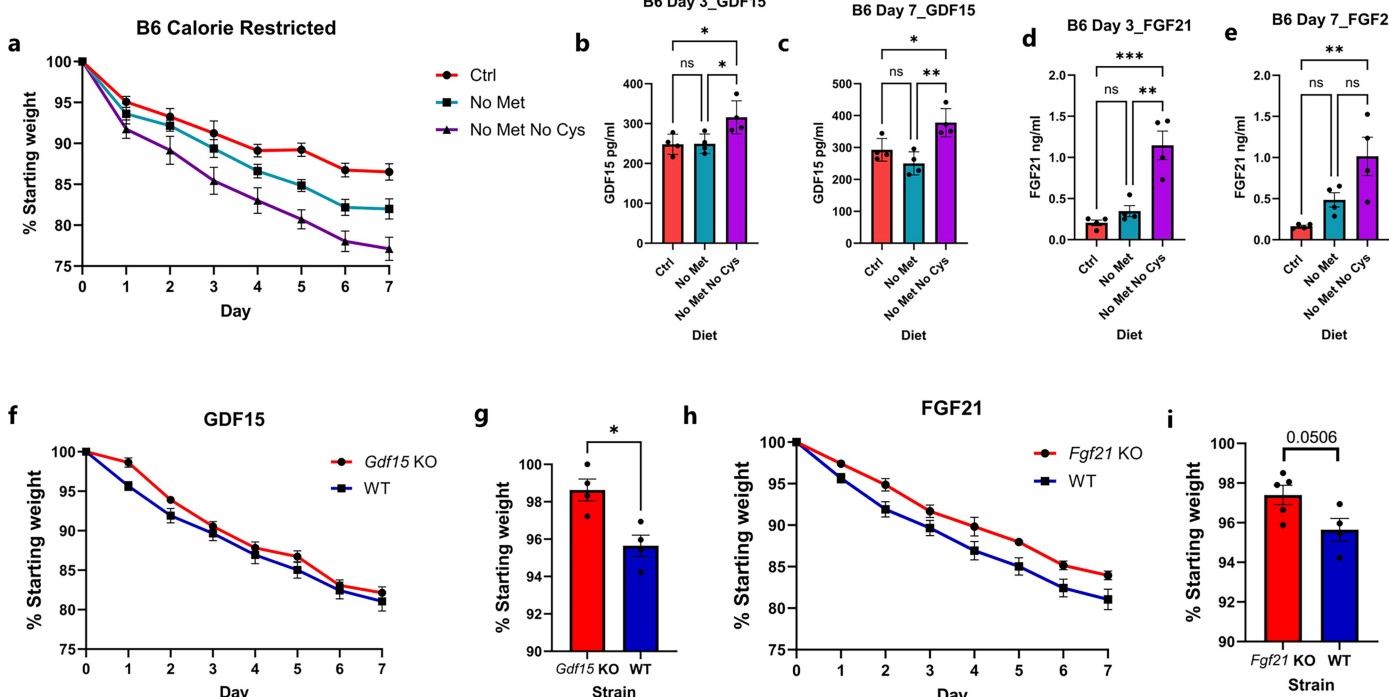

**Extended Data Fig. 5 | Effect of Cysteine and methionine dual restriction on wild type C57BL/6 mice. a**, Weight loss curves of calorie restricted (2.1 g/day) male mice on Ctrl, No Met or No Met No Cys diet (n = 4 per group). **b,c**, Serum GDF15 levels across all three conditions on Day 3 (**b**) or Day 7 (**c**). **d,e**, Serum FGF21 levels across all three conditions on Day 3 (**d**) or Day 7 (**e**) (n = 4 per group). **f**, Weight loss curves of male *Gdf15* KO or WT mice on No Met No Cys diet (n = 4 per group). **g**, Weight after one day of No Met No Cys diet in *Gdf15* KO or WT mice. **h**, Weight loss curves of female *Fgf21* KO or WT mice on No Met No Cys diet (n = 4 for WT and n = 5 for FGF21 KO). **i**, Weight after one day of No Met No Cys diet in *Fgf21* KO or WT mice. Ordinary One-way ANOVA, **b-e**, Unpaired t-test **g, i**, Data are mean ± s.e.m. *P < 0.05, **P < 0.01 and ***P < 0.001.

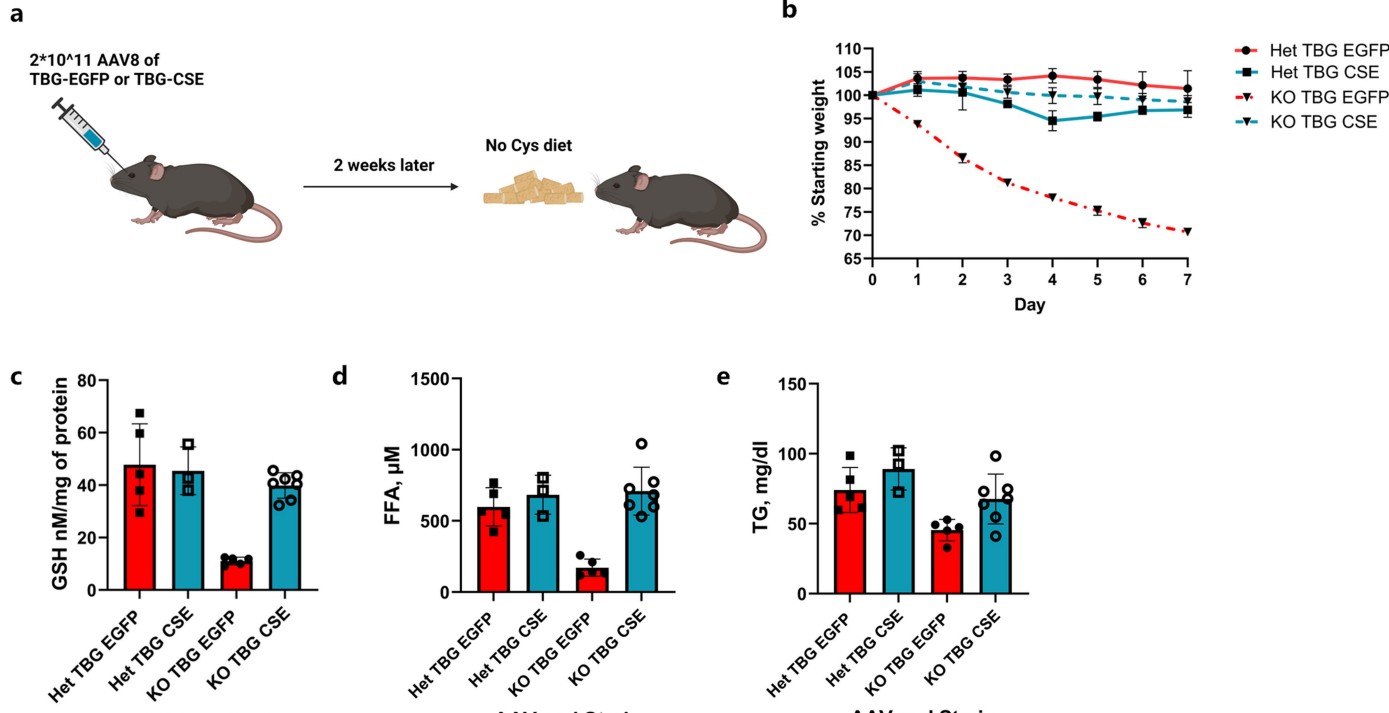

**Extended Data Fig. 6 | Effect of liver specific *Cse* expression on rescuing weight loss in cysteine-deficient mice. a**, Experimental scheme. **b**, Weight loss curves of *Cse* Het or KO mice infected with AAV8-TBG-EGFP or AAV8-TBG-CSE on a No Cys diet. **c-e**, Levels at day 7 of GSH in liver (c) and FFA (d) and TG in serum (e) across all 4 groups (n = 5 Het EGFP, n = 3 Het CSE, n = 5 KO EGFP, n = 7 KO CSE for all). Panel (**a**) was created using BioRender.com. Data are mean ± s.d.

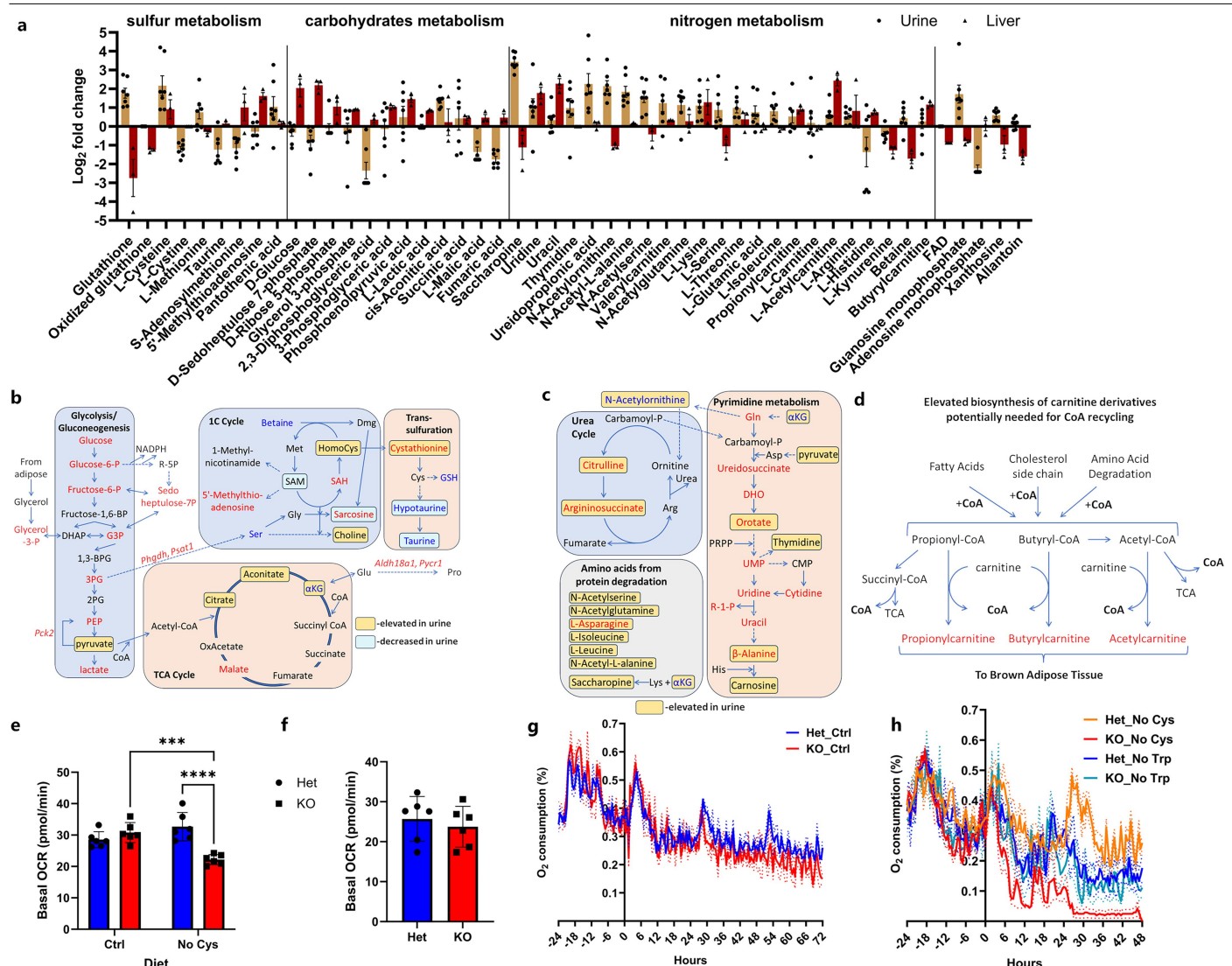

**Extended Data Fig. 7 | Metabolite differences in liver and urine of *Cse* Het and KO mice on No Cys diet. a**, Select metabolites from liver (n = 4 for Het and n = 3 for KO per group) and urine (n = 6 for Het and n = 7 for KO) of KO mice, normalized to levels in Het mice. **b,c**, A cartoon depicting the liver and urine metabolites, that are up- or down-regulated by No Cys diet in KO compared to Het mice, in the context of the major biochemical pathways. Red is higher in KO and Blue is lower. Yellow and blue boxes indicate metabolites that are elevated or reduced, respectively, in urine of KO mice. **d**, Carnitine and CoA metabolism

model to explain excess Acyl-carnitines detected in liver. **e, f**, Seahorse cell energy phenotype assay and basal OCR of CD4 T cells isolated from *Cse* Het and KO mice on Ctrl or No Cys diet (**e**) and No Trp diet (**f**) for 7 days. **g, h**, $O_2$ consumption of *Cse* Het and KO mice restricted to 10% galactose as energy source and fed a control diet (**g**) (n = 4 per group) or a No Cys diet (**h**) compared to a No Trp diet (n = 4 per group) as per scheme in Fig. 5g. Data are mean ± s.e.m. Unpaired t-test **e**, ***P < 0.001 and ****P < 0.0001.

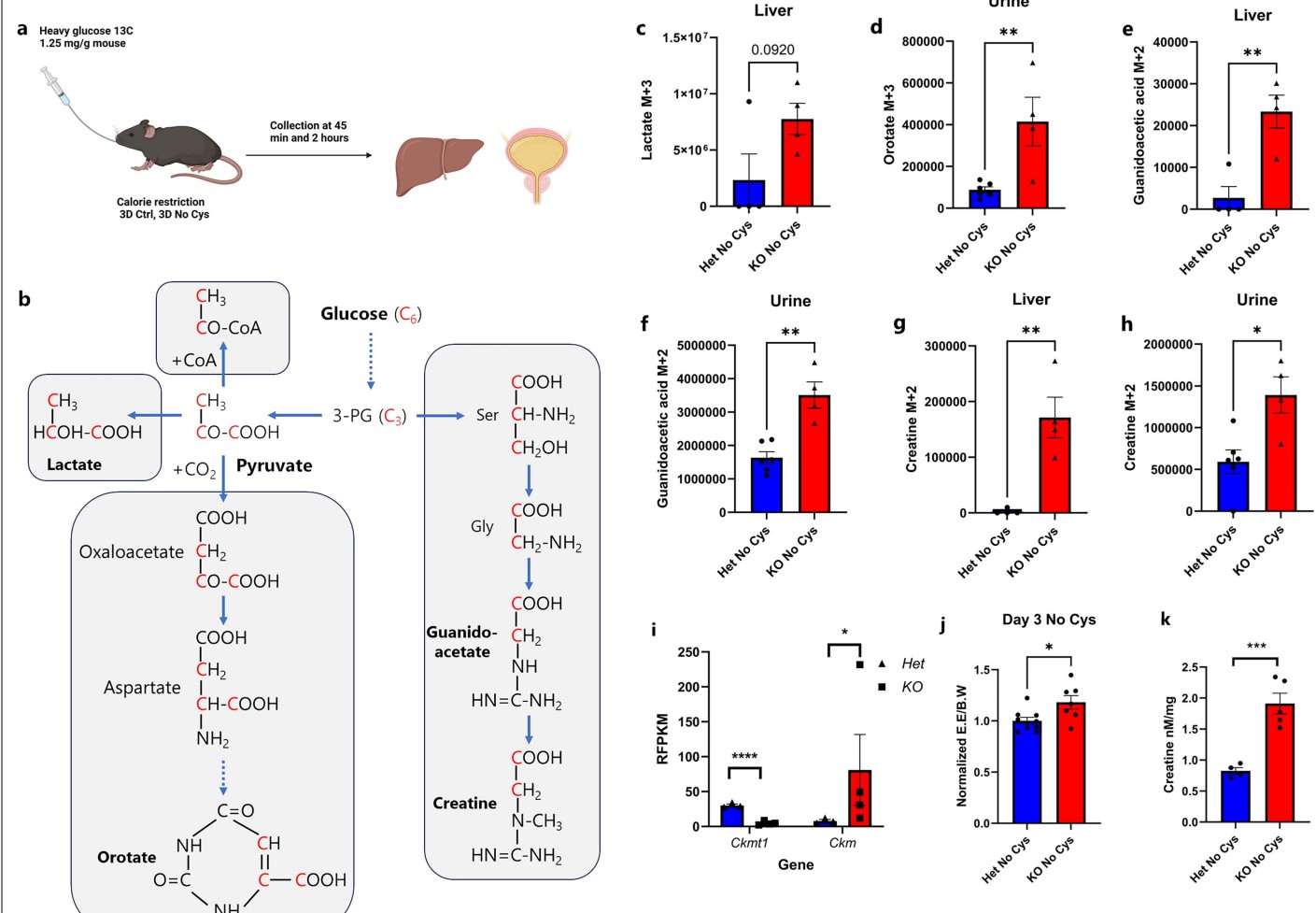

**Extended Data Fig. 8 | ¹³C-Glucose tracing in *Cse* Het and KO mice on Cys-deficient CR diet. a**, Scheme of ¹³C glucose tracing experiment (n = 4 for liver, n = 6 for Het, n = 4 for KO for urine, all males). **b**, Pathway depicting the formation of heavy carbon labelled metabolites from m + 6 glucose. **c**, Levels of m + 3 lactate in the livers of *Cse* Het or KO mice at a time point of 45 min. **d**, Levels of m + 3 orotate in urine of *Cse* Het or KO mice at a time point of 2 h. **e** and **f**, Levels of guanidoacetate m + 2 in liver (e) at 45 min and urine (f) at 2 h in Het or KO mice. **g** and **h**, Levels of creatine m + 2 in liver (g) at 45 min and urine (h) at 2 h in

Het or KO mice. **i**, Expression of *Ckmt1* and *Ckm* in epididymal fat of *Cse* Het and KO mice on No Cys diet (see also Supplementary Table 4) (n = 3 for Het, n = 4 for KO). **j**, Peak energy expenditure (EE) at Day 3 of a No Cys diet normalized to body weight, subsequently normalized to Het data (n = 9 for Het n = 7 for KO). **k**, Creatine levels in liver at Day 6 on a No Cys diet in Het and KO mice (n = 4 for Het, n = 5 for KO). Panel (**a**) was created using BioRender.com. Unpaired t-test **c-k**, Data are mean ± s.e.m. *P < 0.05, ***P < 0.001 and ****P < 0.0001.

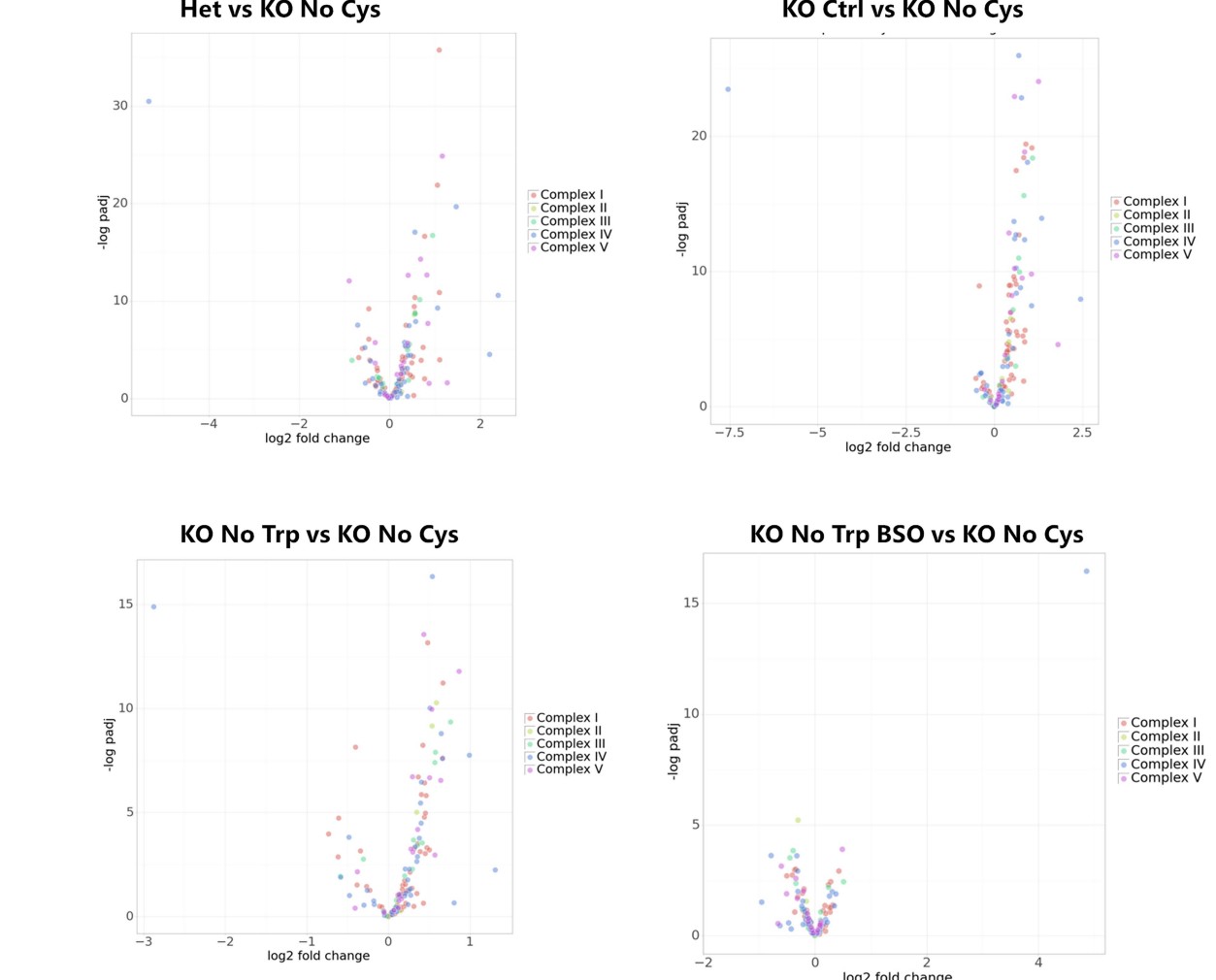

**Extended Data Fig. 9 | Volcano plots of mitochondrial complex genes across both RNA-seq experiments.** Data points are based on liver bulk RNA-Seq data shown in Figs. 3 and 4 (n ≥ 3). Colours represent genes encoding proteins in each of the complexes of the electron transport chain.

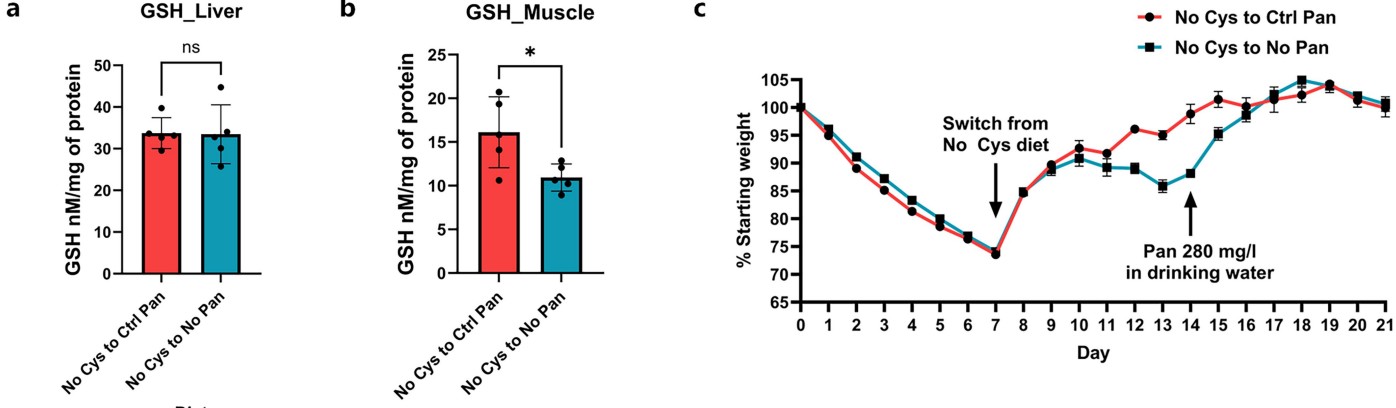

**Extended Data Fig. 10 | Role of CoA in weight recovery. a**, Liver and **b**, Muscle GSH levels at Day 14 of experiment in Fig. 5l (n = 5 per group) **c**, Weight loss curves of female *Cse* KO mice on No Cys diet for 7 days, followed by either 7 days on Ctrl diet or Vit-B5 (Pan) free diet followed by 7 days on a Ctrl diet or a Vit-B5 free diet supplemented with 280 mg/l of Vit-B5 (Pan) (n = 4 per group). Unpaired t-test, Data are mean ± s.d. *P < 0.05.

Evgeny Nudler

# Reporting Summary

## Statistics

For all statistical analyses, confirm that the following items are present in the figure legend, table legend, main text, or Methods section.

| n/a | Confirmed | |
|---|---|---|
| ☐ | ☒ | The exact sample size (*n*) for each experimental group/condition, given as a discrete number and unit of measurement |
| ☐ | ☒ | A statement on whether measurements were taken from distinct samples or whether the same sample was measured repeatedly |
| ☐ | ☒ | The statistical test(s) used AND whether they are one- or two-sided *Only common tests should be described solely by name; describe more complex techniques in the Methods section.* |
| ☐ | ☒ | A description of all covariates tested |
| ☐ | ☒ | A description of any assumptions or corrections, such as tests of normality and adjustment for multiple comparisons |
| ☐ | ☒ | A full description of the statistical parameters including central tendency (e.g. means) or other basic estimates (e.g. regression coefficient) AND variation (e.g. standard deviation) or associated estimates of uncertainty (e.g. confidence intervals) |
| ☐ | ☒ | For null hypothesis testing, the test statistic (e.g. *F*, *t*, *r*) with confidence intervals, effect sizes, degrees of freedom and *P* value noted *Give P values as exact values whenever suitable.* |
| ☒ | ☐ | For Bayesian analysis, information on the choice of priors and Markov chain Monte Carlo settings |
| ☒ | ☐ | For hierarchical and complex designs, identification of the appropriate level for tests and full reporting of outcomes |
| ☐ | ☒ | Estimates of effect sizes (e.g. Cohen's *d*, Pearson's *r*), indicating how they were calculated |

*Our web collection on statistics for biologists contains articles on many of the points above.*

## Software and code

Policy information about availability of computer code

| | |
|---|---|
| Data collection | DEXA Insight, Chemi Doc |
| Data analysis | Prism (GraphPad, v10.0.2), ImageJ |

For manuscripts utilizing custom algorithms or software that are central to the research but not yet described in published literature, software must be made available to editors and reviewers. We strongly encourage code deposition in a community repository (e.g. GitHub). See the Nature Portfolio guidelines for submitting code & software for further information.

## Data

Policy information about availability of data

All manuscripts must include a data availability statement. This statement should provide the following information, where applicable:
- Accession codes, unique identifiers, or web links for publicly available datasets
- A description of any restrictions on data availability
- For clinical datasets or third party data, please ensure that the statement adheres to our policy

Source data will be provided with this paper. Accession codes for sequencing data and metabolomics data are provided. All other data supporting the findings of this study are available from the corresponding authors on reasonable request.

# Research involving human participants, their data, or biological material

Policy information about studies with human participants or human data. See also policy information about sex, gender (identity/presentation), and sexual orientation and race, ethnicity and racism.

| | |
|---|---|
| Reporting on sex and gender | N/A |
| Reporting on race, ethnicity, or other socially relevant groupings | N/A |
| Population characteristics | N/A |
| Recruitment | N/A |
| Ethics oversight | N/A |

Note that full information on the approval of the study protocol must also be provided in the manuscript.

# Field-specific reporting

Please select the one below that is the best fit for your research. If you are not sure, read the appropriate sections before making your selection.

☒ Life sciences ☐ Behavioural & social sciences ☐ Ecological, evolutionary & environmental sciences

For a reference copy of the document with all sections, see nature.com/documents/nr-reporting-summary-flat.pdf

# Life sciences study design

All studies must disclose on these points even when the disclosure is negative.

| | |
|---|---|
| Sample size | Three or more mice per group were used in each experiment. The precise number of animals used are given in the figure legend. The sample size was determined from our previous experience and from what is accepted in the field. Power analysis was done to confirm numbers used were sufficient. |
| Data exclusions | No samples were excluded from the analysis. |
| Replication | Experiments were replicated at least two times. |
| Randomization | Allocation into sample groups was random. In addition, experimental and control mice were used from the same litter whenever possible. Both males and females were used. |
| Blinding | A.V. performed majority of mouse experiments and gave samples to I.G. in a blinded manner for metabolite extraction and analysis. |

# Reporting for specific materials, systems and methods

We require information from authors about some types of materials, experimental systems and methods used in many studies. Here, indicate whether each material, system or method listed is relevant to your study. If you are not sure if a list item applies to your research, read the appropriate section before selecting a response.

## Materials & experimental systems

| n/a | Involved in the study |
|---|---|
| ☐ | ☒ Antibodies |
| ☒ | ☐ Eukaryotic cell lines |
| ☒ | ☐ Palaeontology and archaeology |
| ☐ | ☒ Animals and other organisms |
| ☒ | ☐ Clinical data |
| ☒ | ☐ Dual use research of concern |
| ☒ | ☐ Plants |

## Methods

| n/a | Involved in the study |
|---|---|
| ☒ | ☐ ChIP-seq |
| ☒ | ☐ Flow cytometry |
| ☒ | ☐ MRI-based neuroimaging |

## Antibodies

| | |
|---|---|
| Antibodies used | anti-Phospho-eIF2α (Cell Signaling Technology 3597), anti-eIF2α (Cell Signaling Technology 2103) anti-Phospho-ACC (Cell Signaling Technology 3661), anti-ACC (Cell Signaling Technology 3662), anti-βTubulin (Proteintech 10068-1-AP), Anti-β-Actin−Peroxidase |

antibody (Sigma A3854), UCP1 (CST, Cat#: 72298S, Clone: E9Z2V ), CASP3 (CST, Cat#: 9579S, Clone: D3E9 ), NQO1 (Sigma-Aldrich, HPA007308), Custom NRF2 (1:1000, provided by Edward Schmidt, Montana State University)

Validation | All Antibodies except anti-NRF2 were used according to recommendations of the manufacturer. Anti-NRF2 antibody usage was optimized using Keap1 mutant (Nrf2 overproducing) and Nrf2 KO tumor cell lines.

# Animals and other research organisms

Policy information about studies involving animals; ARRIVE guidelines recommended for reporting animal research, and Sex and Gender in Research

Laboratory animals | Mice were bred and maintained in the Alexandria Center for the Life Sciences animal facility of the New York University School of Medicine, in specific pathogen-free conditions. C57BL/6 mice (Jax 000664),  Gcn2 KO (B6.129S6-Eif2ak4tm1.2Dron/J) and Fgf21 KO (B6.129Sv(Cg)-Fgf21tm1.1Djm/J)  were purchased from Jackson Laboratories. Cse KO (129/C57BL/6 background) were generated by Rui Wang as previously described and given to us by Christopher Hine.Gdf15 KO mice were generated by Eileen White's lab and is described in the methods section. Mice in all the experiments were at least 9 weeks old at the starting point of various diets unless described specifically.

Wild animals | N/A

Reporting on sex | Both males and females were used for experiments. Most weight curves shown are exclusively from male mice or female mice as there is a small variation in weight loss on day 1. To reduce variation during RNA-seq and metabolomics, only male mice were used.

Field-collected samples | N/A

Ethics oversight | All animal procedures were performed in accordance with protocols approved by the Institutional Animal Care and Usage Committee of New York University School of Medicine.

Note that full information on the approval of the study protocol must also be provided in the manuscript.

# Plants

Seed stocks | N/A

Novel plant genotypes | N/A

Authentication | N/A

