## [Peer Review File · Nature]

Unraveling cysteine deficiency-associated rapid weight loss

Corresponding Author: Professor Evgeny Nudler

Version 0:

Reviewer comments:

Referee #1

(Remarks to the Author)

The manuscript has investigated the effects of different diets on weight loss and found that on a CST KO background a low cysteine diet causes significant weight loss. While initially quite excited by this finding I found the data to not robustly support the conclusions. Further there is no mechanism to support the findings other than correlative RNAseq analysis. In general, I found the paper while interesting to be quite preliminary in nature.

1. Figure 1. All data presented is as percent changes in an extremely low number of animals per group (n=2-4). It is essential authors present absolute body mass changes. How many times were experiments repeated?. The sex and age of the animals was also not provided. In general there are serious concerns that while the data are interesting and may be suitable for a pilot, it must be reproduced in a larger group of animals to see if it is reproducible and absolute body mass data presented.
2. Figure 2D RER data said to be different but no statistics provided indicating this is the case? An n=4 animals was studied in metabolic cages but then there are only 3 animals presented for body composition analysis and again everything is relative a percentage. What were the starting body mass of the animals. Were they males, females, ages are not presented? In general as above the sample sizes for these data is considered extremely small and it is very surprising that there are significant differences with such a low n.
3. The authors propose the potential importance of GDF15 and ACC in mediating the effects but no mechanistic studies probing whether this is actually the case are completed. As these data are only correlative in nature and there are likely many things changing with the diets it is essential a much more detailed studies be completed.

Minor Comments:

1. Figure 5. It is not clear what the relevance to weight loss is to the phenotypes observed.

Referee #2

(Remarks to the Author)

This is a fascinating paper in which the researchers have systematically removed individual amino acids and studied their impact on weight loss. Remarkably, they observed that inducing conditional cysteine restriction led to the most significant weight loss, with a 20% reduction within 3 days and a 30% reduction within one week. What's even more impressive is that this effect could be easily reversed. The results are truly striking! Interestingly, it appears that the mechanism behind this weight loss is not primarily due to a decrease in glutathione (GSH), but rather stems from an increase in the utilization of fat mass. Mechanistically, the researchers have linked cysteine deficiency to the activation of the integrated stress response (ISR) and NRF2-mediated oxidative stress response (OSR), which in turn leads to the induction of GDF15. It's worth noting that GDF15 is well-known for its role in increasing lipolysis and causing food aversion. I find the observations in this paper quite intriguing and super cool, but it's important for the authors to address some apparent gaps in the mechanistic links they've identified.

Major experiments that need to be addressed include:

1. Observation of a robust ISR response in the liver, including GDF15 and FGF21, raises the question of whether liver deficiency of CSE is a crucial factor driving weight loss. To investigate this, researchers should employ AAV strategies for delivering CSE cDNA to the liver through intravenous (I.V.) injections, which is a standard approach in the field. This would

result in a mouse model with CSE deficiency everywhere except in the liver. Does this rescue the phenotype?

2. An evident experiment is to determine the necessity of GDF15 and FGF21 in causing weight loss. Researchers should inhibit GDF15 and FGF21 by administering neutralizing antibodies for both (as described in PMID: 31402172). There are well-characterized antibodies available for both hormones.

3. Another apparent experiment is to assess the importance of ISR by using the well-established ISRIB compound. This experiment would involve examining tissue ISR responses using RNA seq in conjunction with weight loss.

It is essential to emphasize that these experiments do not require the breeding of new mice; instead, they involve straightforward interventions that can provide valuable mechanistic insights into causality. The goal is to determine whether ISR and GDF15 are indeed necessary factors, regardless of the outcome.

Referee #3

(Remarks to the Author)

In this manuscript by Varghese et al, the authors found that cysteine deficiency, using a combination of cysteine-deficient diet and transsulfuration pathway-deficient mouse model, induced rapid and reversible weight loss. They meticulously characterized the physiological parameters under cysteine deficiency and found that browning cysteine deficiency causes reduced food intake and rapid browning of adipose tissues. Mechanistically, cysteine deficiency leads to simultaneous activation of the integrated stress response and oxidative stress response and the expression of the anorexic hormone GDF15. Additionally, they found that cysteine deficiency led to the reduction of CoA levels in liver and muscle, causing disruption of metabolism reliant on CoA-containing intermediates.

Collectively, this is a well-designed study that offers a lot of useful insight to the rewiring of systemic metabolism under amino acid depletion. My major concern with this manuscript is its novelty: activation of ISR/OSR by cysteine depletion has been demonstrated in many cellular, physiological and pathological scenarios, and the role of GDF15 in mediating systemic response to ISR is well-documented. Depletion of CoA, a major downstream fate of cysteine, by cysteine depletion, is also expected, and the authors provided limited data on the link between CoA depletion and systemic metabolic effect. Additionally, reliant on Cse-KO animals, though understandably an appropriate tool for inducing cysteine depletion, may nevertheless limit the physiological relevance of the study. Provided that the authors address the specific points below, I would recommend this manuscript for a specialized journal such as Nature Metabolism; for it to be considered by Nature, I would expect other findings to be incorporated into the manuscript that substantially boost its novelty.

It seems counterintuitive that cysteine depletion leads to substantially decrease in Coenzyme A and switch towards more utilization of lipid, as shown by the RER. Could the authors provide an explanation for this?

The rapid browning of white fat in 1H is quite striking. Is there data on total energy expenditure that quantitatively measures the contribution of browning on weight loss?

The author implies that cysteine loss, compared to tryptophan loss, leads to more activated ISR due to its low levels and high demands. It would be very helpful if the author could justify this claim by measuring the decrease in cysteine/tryptophan upon dietary depletion (for measuring cysteine this may require derivatization). The levels of activation of ISR is also related to the Km of the respective aminoacyl-tRNA synthetase; the authors could also consider discussing this in their manuscript.

Compared to NoTry+BSO, NoCys diet seems to induce more weight loss only after 5 days. However, most transcriptomics/metabolomics measurements were performed on day 2. Could the authors provide a justification for choosing this time point? BSO is a relatively weak inhibitor and prolonged treatment may lead to a compensatory increase in the expression of genes in GSH synthesis such as SLC7A11, GCLC, GCLM, which may dampen the effect of drug treatment at the later time points.

In characterizing the metabolic inefficiency caused by CoA deficiency, the authors used many metabolite levels in tissue versus urine as a proxy for metabolic flux; this is a very indirect snapshot of the metabolic flux. If possible, the author could consider in vivo tracing with isotope-labeled metabolite to have better quantification of metabolic inefficiency.

Additionally, the contribution of metabolic inefficiency to weight loss is unclear. (How) Does it contribute to increased lipolysis in the adipose tissue? How much energy is lost via excretion as urinal metabolites?

Cysteine affects OXPHOS via multiple pathways (such as CoA, GSH and Fe-S cluster synthesis), so defective OXPHOS is very weak evidence that CoA is mediating the effect of cysteine depletion. The authors should try to clarify the contribution of CoA in mediating OXPHOS defects.

Version 1:

Reviewer comments:

Referee #1

(Remarks to the Author)

I have reviewed the response to reviewers document and the revised manuscript. While the authors have addressed some of my concerns with respect to mechanisms I do not believe they have adequately addressed my concern about the very limited sample sizes for many experiments which lay the foundation for the paper. Specifically, it is stated in the response to reviewers that "All experiments presented in the revised manuscript, except for the metabolomics and sequencing, were conducted at least twice with a minimum of three replicates per group". From my understanding of the above statement then there should be a minimum of a n=6 per group, which would seem a reasonable number of mice to avoid a type 2 error and ensure reproducibility. However, from data presented in Figures 1-2 and Extended Data Figures 1-3 it is clear this is not the case as most experiments have an n=2-4. Moreover the sample size in many of the new experiments are also below this threshold (i.e. figure 4,5,6, extended data figure 6a,b and 7 all n=3-4.

Referee #2

(Remarks to the Author)

The authors have done a good job of responding.

Referee #3

(Remarks to the Author)

The authors have added evidence that significantly strengthened the observations in the original submission. In particular I'm pleased to see the inclusion of the B5 diet experiment to pinpoint the role of CoA in mediating weight loss as well as tracing with labeled glucose to better characterize the metabolic inefficiency caused by CoA depletion. Overall the revised manuscript is much more comprehensive in characterizing the metabolic phenotypes and offered more mechanistic explanations. However the how (much) exactly ISR/OSR and "metabolic efficiency" contribute to rapid weight loss still lacks convincing explanation, possibly due to limitations in currently available technologies.

Version 2:

Reviewer comments:

Referee #1

(Remarks to the Author)

Version 4:

Reviewer comments:

Referee #1

(Remarks to the Author)

Referee #3

(Remarks to the Author)

The data in Figures 1 and 2 exhibit large effect sizes, making the current experiments acceptable. However, I agree with the other reviewer that using only three mice per group is uncommon, as most studies typically include a minimum of five to six mice. Given the robustness of the data, I find these acceptable.

For Figure 2 (and several other heat-maps), the RNA-seq analysis lacks statistical validation. With only three mice per group, some selected genes may not be significantly altered (see Fig. 3b). The authors should account for this when presenting their gene expression data. Otherwise it is a bit misleading. (perhaps indicate next to the heat-map or show only those genes that are significant)

My main concern lies with Figures 5c and 5n. The authors attribute their phenotype to CoA depletion, yet the effect sizes in these panels are small, and the statistical support appears weak. They should either soften their claims regarding CoA or provide a third biological replicate to strengthen their conclusions.

We are grateful for all the reviewers' comments and queries, which have helped us to significantly improve our manuscript by providing more mechanistic insights. We have made substantial changes to the text, updated and added several new panels to main Figures 4 and 5, included five new Extended Data figures and one new supplementary table. In this letter, we have included five "Reviewer Figures" and one "Reviewer Video", to address various queries raised by the reviewers. We would be happy to include any or all of these data in the manuscript if deemed necessary. We hope that these experiments address all the reviewers' questions and concerns and strengthen our manuscript.

Also, there is another manuscript from the lab of Vishwa Deep Dixit under consideration at the journal (see attached their 2nd revision as Supplemental File 1, referred to as Lee et al) which also primarily describes the rapid weight loss of Cse KO mice on a cysteine free diet. We have several identical observations, although they have focused and conducted extensive work on fat loss and thermogenesis, rather than metabolic pathways that we have investigated. This additional manuscript highlights the strong reproducibility of our phenotype as they use a different Cse KO with a different cysteine-free diet in a different facility.

REFEREE #1:

The manuscript has investigated the effects of different diets on weight loss and found that on a CST KO background a low cysteine diet causes significant weight loss. While initially quite excited by this finding I found the data to not robustly support the conclusions. Further there is no mechanism to support the findings other than correlative RNAseq analysis. In general, I found the paper while interesting to be quite preliminary in nature.

We appreciate the concerns raised by the reviewer and have provided additional extensive experimental data to improve the mechanistic understanding of our study, as described below.

1) Role of CoA in weight loss and metabolic inefficiency

CoA is indispensable for several critical metabolic processes, including the conversion of pyruvate to Acetyl CoA, normal progression of the TCA cycle, fatty acid (FA) synthesis and FA oxidation etc. CoA deficiency forces mice to rely predominantly on glycolysis for energy production, minimize the use of the TCA cycle and mitochondria and decreases FA and TG biosynthesis. Consistent with this hypothesis, we observed an increase in steady-state levels of glycolysis intermediates (**Fig. 5f**) and low TG levels (**new Fig. 4h**) in the circulation of Cse^{-/-} mice on Cys-free diet. Furthermore, these mice cannot survive on galactose as their sole carbon source (**Fig. 5g**), since galactose metabolism does not produce net ATP via glycolysis.

Previous experiments with a Vitamin B5 (pantothenic acid)-deficient diet demonstrated no loss of CoA over a two-month period (PMID: 3807784). However, in the adipose and muscle tissues of *Cse*^{-/-} mice on Cys-free diet, we detected upregulation of *Vnn1* and *Vnn3* genes, implying that CoA is degraded to B5 and cysteamine. Accordingly, we observed a decrease in CoA concentration in multiple tissues (**Fig 5b-d**).

To determine the role of CoA, we noted that during the phase of No Cys, the mice also lost pantothenic acid (vitamin B5) in urine, since B5 is a water-soluble vitamin that cannot accumulate, and CoA resynthesis is limited due to lack of cysteine (**Fig 5i, j**). By Day 7, along with the depletion of other cysteine-containing molecules, B5 levels would also be reduced across tissues. We hypothesized that if we restore cysteine, but without Vit B5, after 7 days of No Cys, we could rescue all cysteine-containing molecules except CoA, which would be limited to the remaining tissue B5 levels or what was absorbed that day (**new Fig 5k**).

Indeed, *Cse*^{-/-} mice previously fed a Cys-free diet failed to gain as much weight when subsequently fed a B5-deficient Cys-sufficient diet compared to a diet sufficient in both (**new Fig. 5l**). While we detected rescue of GSH, CoA levels remained significantly lower on the B5-deficient diet (**new Fig 5m, n and new Extended Data Fig 13a, b**). Furthermore, administering B5 in drinking water after 7 days on the B5-deficient diet rapidly rescued the weight recovery difference (**new Extended Data Fig 13c**).

Taken together, our experiments clearly demonstrate that:

- (i) CoA is quickly and progressively lost on a Cys-free diet.
 - (ii) Cys, rather than B5 availability, is a key determinant of cellular CoA levels.
 - (iii) CoA loss shifts cells to glycolysis, accelerates the wasting of intermediary metabolites in urine, decreases mitochondrial use, and reduces TG production.
- These collectively contribute to the rapid weight loss observed in *Cse*^{-/-} mice on a Cys-free diet.

2) Role of GCN2 and GDF15 in weight loss and feeding

Both GCN2 activation in the brain and GDF15 production have been shown to drive food aversion (PMID: 15774759, PMID: 31875646). GCN2 senses amino acid deficiency to activate the integrated stress response (ISR). To test the contribution of each to weight loss, we decided to delete or inhibit GCN2 or GDF15.

a) Knockout or inhibition of GCN2

We created a double KO of *Gcn2* and *Cse* but found that by 8 weeks nearly all *Gcn2* *Cse* DKO mice developed hindlimb paralysis and had to be euthanized (**Reviewer Video 1**). As both single *Cse* or *Gcn2* KO mice are viable, this result strongly indicates that GCN2 is required and activated in *Cse* KO mice to compensate for Cys deficiency due to the lack of the transsulfuration pathway. GCN2 likely increases the import of dietary Cys to ensure normal growth in *Cse* KO mice.

As an alternative to *Gcn2* KO, we used a commercially available GCN2 inhibitor, GCN2iB, which did not change the weight loss kinetics in the *Cse* KO (PMID: 30061420). However, at an 8 mg/kg dose, GCN2iB induced significant weight loss in *Cse* het mice on a No Cys diet (**Reviewer Fig 1A and B**). We also did not observe any changes in food intake in *Cse* KO mice (**Reviewer Fig 1C**). However, we detected a substantial increase in GDF15 levels in the serum of *Cse* KO mice, which explains why their food intake remained low, thus not affecting weight loss kinetics (**Reviewer Fig 1D**).

The observed weight loss in Hets given GCN2iB, suggests that inhibiting the response to No Cys diet, even in the presence of *Cse*, could lead to changes resulting in weight loss (**Reviewer Fig 1B**).

Taken together, these results indicate that GCN2 activation is required to compensate for Cys deficiency and likely prevents weight loss in the absence of cysteine in hets.

Additionally, it reveals that there are alternative pathways to activate the response to low Cys stress (GDF15 and others), which could negate any expected weight loss or phenotypic changes in the absence of GCN2.

Reviewer Figure 1: Effect of GCN2 inhibitor on weight loss. **A)** Weight loss curves of *Cse* Het and KO on No Cys with 8 mg/kg of GCN2iB given i.p. or vehicle ctrl. **B)** Weight loss curves (conditions as in A) for hets. **C)** Daily food intake in *Cse* KO with 8 mg/kg of GCN2iB or vehicle ctrl. **D)** Serum GDF15 level in *Cse* KO with 8 mg/kg of GCN2iB or vehicle ctrl on Day 3.

b) Anti-GDF15 blocking antibody

We administered 20 mg/kg of anti-GDF15 or control anti-KLH antibodies i.p. to Cse KO mice 2 hours before shifting them to a No Cys diet. Despite this being twice the dose used in previous studies, we observed no difference in either food intake or weight loss patterns (**Reviewer Fig 2 A, B**) (PMID: 31402172, PMID: 37437602).

There are multiple explanations for the observed results. Previously, this antibody was used at 10 mg/kg to study one-time acute stress responses such as LPS injection or allergy response (PMID: 31402172, PMID: 37437602). In our experiment, we have a chronic cysteine depletion, which could, over time, induce a compensatory response upregulating GDF15. Alternatively, this particular batch of antibody might not be efficient in our system.

Reviewer Figure 2: Effect of anti-GDF15 antibody on weight loss. A) Weight loss curves of Cse KO on No Cys with single dose 20 mg/kg of anti-GDF15 antibody or ctrl anti-KLH antibody given i.p. 2 hours prior to switching to No Cys diet. **B)** Daily food intake in Cse KO with anti-GDF15 or anti-KLH antibody.

3) Role of Integrated Stress Response

To test the role of ISR in weight loss, we gave ISR inhibitor (ISRIB) daily to Cse KO mice on a No Cys diet (PMID: 33258451). We detected a small but statistically significant reduction in weight loss on Day 1 (**Reviewer Fig 3A and B**). However, by Day 7 the ISRIB-treated mice lost the same amount of weight as untreated controls (**Reviewer Fig 3A**). ISRIB partially suppressed the phosphorylation of eIF2a (**Reviewer Fig 3C**). Although, as expected, the levels of GDF15 and FGF21 were downregulated (**Reviewer Fig 3D and E**) (PMID: 34877504), they were still significantly above the baseline level in WT animals (~ 300 pg/ml GDF15 and ~0.2 ng/ml FGF21, **New Extended Data Figure 7c and e**).

These results correlate with the previously published data demonstrating that there are *P*-eIF2a-dependent and -independent pathways that converge to upregulate typical ISR transcription in response to sulfur amino acid deficiency (PMID: 27613409, PMID: 28446632). Moreover, a diet low in Cys and Met induces expression of “classical” ISR

genes even in *Atf4*^{-/-} mice (PMID: 33512502). Together, this suggests that compensatory pathway(s) are induced by Cys deprivation and can initiate an ISR signature transcription critical for survival on a Cys-free diet.

Reviewer Figure 3: Effect of ISRIB on weight loss. **A)** Weight loss curves of Cse KO on No Cys with daily dose 1 mg/kg ISRIB or 2.5 mg/kg ISRIB or vehicle ctrl. **B)** % starting weight across three conditions on Day 1. **C)** % phosphorylation of eIF2a across three conditions in the liver on Day 7. **D)** Serum GDF15 on Day 7. **E)** Serum FGF21 levels on Day 7.

We also respectfully disagree with the statement that we only used correlative RNA-seq analysis to make our conclusions. The No Trp+BSO experiments, combined with various biochemical measurements, demonstrated how different stresses induced by cysteine deprivation affect weight loss and how individual stresses do not trigger similar responses (Figure 4). Also, we have now included new data on the level of triglycerides (TG) and free fatty acids (FFA) in serum, which supports our RNA-seq results (**new Fig. 4g and h, new Extended Data Fig 8d and e**).

Thus, we feel that we have significantly strengthened the manuscript with the new experimental data and the improved description of the implications of the results.

1. Figure 1. All data presented is as percent changes in an extremely low number of animals per group (n=2-4). It is essential authors present absolute body mass changes.

How many times were experiments repeated?. The sex and age of the animals was also not provided. In general there are serious concerns that while the data are interesting and may be suitable for a pilot, it must be reproduced in a larger group of animals to see if it is reproducible and absolute body mass data presented.

All experiments presented in the revised manuscript, except for the metabolomics and sequencing, were conducted at least twice with a minimum of three replicates per group either as a standalone experiment or had the groups repeat across different experiments. The phenotype as one can easily observe from all the weight curves or even sequencing data (Het and KO in No Cys in Fig 3 and Fig 4) is quite consistent across different experiments, showcasing its reproducibility.

Furthermore, Lee et al in their manuscript (see supplemental file 1), have reproduced multiple experiments in a different facility with a different Cse KO mouse with a different cysteine-free diet. This strongly highlights the reproducibility of the model and our data.

As suggested by the reviewer, we have included figures showing the absolute and percent body weight change side by side. Mice, even from the same strains and litters, exhibited a 10-15% body weight variation, which increases the noise in our analysis when using absolute body weight. Also, we have included a graph plotting percentage body weight change against the starting weights of Cse KO mice on a No Cys diet to demonstrate that these parameters are independent (**new Extended Data Fig 2**).

In the revised version, we have updated the text to indicate the biological sex if only one biological sex was used. In the Methods section, we now specify that mice used for all experiments were between 10-14 weeks old. Both male and female mice display the same weight loss phenotype, with female mice showing slightly lower weight loss on Day 1 (**new Extended Data Fig 1k**). This difference from Day 1 persists, so we decided against pooling data from both males and females to avoid increasing the noise in the data.

For sequencing and metabolomics, we used male mice to avoid the noise due to inherent differences in male and female mice, which would hinder our ability to identify relevant differences.

2. Figure 2D RER data said to be different but no statistics provided indicating this is the case? An n=4 animals was studied in metabolic cages but then there are only 3 animals presented for body composition analysis and again everything is relative a percentage. What were the starting body mass of the animals. Were they males, females, ages are not presented? In general as above the sample sizes for these data is considered extremely small and it is very surprising that there are significant differences with such a low n.

As suggested, we have updated the main manuscript and figures to indicate the number of animals used and specify their sex.

We used 4 animals per group for the body composition analysis using DEXA. We believe that 4 animals per group is sufficient to achieve significance due to the dramatic decrease in body fat observed.

For the RER graph, we apologize for not highlighting the time points that were statistically significant. We have now indicated the time points with significant differences (**Revised Fig 2d**).

3. The authors propose the potential importance of GDF15 and ACC in mediating the effects but no mechanistic studies probing whether this is actually the case are completed. As these data are only correlative in nature and there are likely many things changing with the diets it is essential a much more detailed studies be completed.

For GDF15, as we explained above in **Reviewer Fig 2**, we could not achieve a rescue with the GDF15 blocking antibody in the *Cse* KO. However, in our experiments with WT B6 mice on a No Met No Cys diet, we again saw a significant increase in GDF15 and FGF21 (**new Extended Data Fig 7a-e**). In the absence of available *Gdf15* *Cse* DKO mice, we used *Gdf15* KO mice on a No Met No Cys diet to test if even a slight increase in GDF15 (in B6 [300pg/ml] compared to *Cse* KO on No Cys [800 pg/ml]) could play a role. We indeed observe a significant decrease in weight loss on Day 1 in the No Met No Cys diet in the *Gdf15* KO compared to B6 WT mice. The differences did not increase by Day 7, likely driven by GCN2 activation compensating for the lack of GDF15, an inverse of what we showed in Reviewer Fig 1 (**new Extended Data Fig 7f and g**).

For ACC1, our WB analysis demonstrated a threefold reduction of the protein level in the liver of *Cse*^{-/-} mice on a No Cys diet (**revised Fig 4f**). As we did not find any changes in *Acaca* gene expression but detected strong upregulation of *Trib3*, it is likely that ACC1 is downregulated by ubiquitin-dependent degradation (PMID: 16794074). We believe that genetic rescue experiment would require the generation of *Acaca* gene mutant mice with potentially multiple point mutations, which is beyond the scope of this work.

As liver ACC1 is the key enzyme controlling *de novo* lipogenesis, we measured circulating TG. Consistent with ACC1 degradation, the level of TG was significantly reduced in the *Cse* KO compared to Het on a No Cys diet (**new Fig 4h**). Despite increased fat loss from adipocytes, as shown in **Fig 2f**, together with the downregulation of cholesterol biosynthetic genes (**Fig. 3c and 4k**) and ACC1 degradation, this strongly suggests a decreased level of lipid particle production by the liver, contributing to weight loss.

Our new results with AAV restoring *Cse* expression in the liver show that expression of *Cse* in the liver specifically can restore TG and FFA levels to those seen in control animals (**new Extended Data Fig 8**)

Hence, we only have supporting evidence for the role of GDF15 and ACC1 in weight loss. We have adjusted the tone in the revised manuscript to reflect this reality.

Minor Comments:

1. Figure 5. It is not clear what the relevance to weight loss is to the phenotypes observed.

We thank the reviewer for pointing out that we made Figure 5 and its relevance hard to interpret and understand. We have updated the text around the figure and the scheme in Figure 5a to make it more accessible.

Briefly, Fig 5 explains how Cys deficiency causes progressive loss of CoA, which in turn leads to metabolic inefficiency, loss of carbohydrates and intermediary metabolites in urine, and causes mice to rely heavily on glycolysis rather than oxidative phosphorylation for energy production.

When we compared animals on No Cys and No Trp + BSO diets, we observed that despite similar ISR, OSR, GDF15, ACC levels and transcriptional responses, there was greater weight loss in the No Cys condition (**revised Fig. 4**). To explain the remaining weight loss not accounted for by oxidative stress response (OSR) and integrated stress response (ISR), we investigated other molecules affected by a No cysteine diet. Given that CoA biosynthesis requires cysteine and that we observed an increase in CoA degradation machinery, we measured its level. CoA degradation to pantothenate is critical for inter-organ transport of CoA. However, upon degradation by VNN1 (vanin) the thiol moiety of CoA is irreversibly lost as cysteamine (**revised Fig. 5a**). The resynthesis of CoA requires supply of new Cys, which is obviously limited in a Cse KO mice on a No Cys diet. Cumulatively, this results in pantothenate loss in the urine and significantly decreased tissue CoA levels on the No Cys diet (**revised Fig. 5b-d, j**).

A loss of CoA would slow down the conversion of pyruvate to Acetyl-CoA and alpha-ketoglutarate to Succinyl-CoA, ultimately inhibiting the TCA cycle and mitochondrial respiration and inducing loss of pyruvate, citrate, a-ketoglutarate, and their derivatives and precursors in urine (**Extended Data Fig 9b**). As a result, we proposed that Cse KO mice on a No Cys diet progressively rely on glycolysis for energy production. Indeed, in **Fig. 5g** we confirmed that Cse KO mice on a No Cys diet cannot survive on galactose (which does not produce net ATP in glycolysis) as a sole energy source (**Fig. 5g**). We further demonstrated at a cellular level (using T-cells) a reduction in oxygen consumption in T cells from Cse KO mice fed a No Cys diet (**revised Extended Data Fig. 11**)

In the revised manuscript and Fig. 5, we also show experiments demonstrating that CoA deficiency plays a significant role in the weight loss phenotype (**New Fig. 5k-n**).

REFeree #2

This is a fascinating paper in which the researchers have systematically removed individual amino acids and studied their impact on weight loss. Remarkably, they observed that inducing conditional cysteine restriction led to the most significant weight loss, with a 20% reduction within 3 days and a 30% reduction within one week. What's even more impressive is that this effect could be easily reversed. The results are truly striking! Interestingly, it appears that the mechanism behind this weight loss is not primarily due to a decrease in glutathione (GSH), but rather stems from an increase in the utilization of fat mass. Mechanistically, the researchers have linked cysteine deficiency to the activation of the integrated stress response (ISR) and NRF2-mediated oxidative stress response (OSR), which in turn leads to the induction of GDF15. It's worth noting that GDF15 is well-known for its role in increasing lipolysis and causing food aversion. I find the observations in this paper quite intriguing and super cool, but it's important for the authors to address some apparent gaps in the mechanistic links they've identified.

We thank the reviewer for their comments, and we have performed the requested experiments as well as additional ones to enhance our understanding of the phenotype.

Major experiments that need to be addressed include:

1. Observation of a robust ISR response in the liver, including GDF15 and FGF21, raises the question of whether liver deficiency of CSE is a crucial factor driving weight loss. To investigate this, researchers should employ AAV strategies for delivering CSE cDNA to the liver through intravenous (I.V.) injections, which is a standard approach in the field. This would result in a mouse model with CSE deficiency everywhere except in the liver. Does this rescue the phenotype?

This was a great suggestion, and we procured an AAV8 vector that expresses mouse *Cse* under a TBG promoter or EGFP under a TBG promoter as a control. We administered 2×10^{11} viral particles per mouse based on doses typically used for liver-specific AAV models (PMID: 34435198, PMID: 30112420). We found that liver-specific expression of *Cse* completely rescued the liver GSH levels and prevented the weight loss (**new Extended Data Fig 8a-c**). We also detected a rescue of TG and FFA levels (**new Extended Data Fig 8d, e**), suggesting that Cys availability in the liver regulates *de novo* lipogenesis and contributes to fat levels.

2. An evident experiment is to determine the necessity of GDF15 and FGF21 in causing weight loss. Researchers should inhibit GDF15 and FGF21 by administering

neutralizing antibodies for both (as described in PMID: 31402172). There are well-characterized antibodies available for both hormones.

Although the suggested experiments appear straightforward, the results that we obtained were challenging to interpret. We administered 20 mg/kg of anti-GDF15 or control anti-KLH antibodies i.p. to Cse KO mice 2 hours before shifting them to a No Cys diet. Despite this being twice the dose used in previous studies, we observed no difference in either food intake or weight loss patterns (**Reviewer Fig 2A, B**) (PMID: 31402172, PMID: 37437602).

There are several potential explanations for these results. Previously, this antibody was used at 10 mg/kg to study acute stress responses, such as LPS injection or allergy responses (PMID: 31402172, PMID: 37437602). In our experiment, we dealt with chronic cysteine depletion, which could over time induce a compensatory response upregulating GDF15. Alternatively, this particular batch of antibody might not be efficient in our system.

Reviewer Figure 2: Effect of anti-GDF15 antibody on weight loss. A) Weight loss curves of Cse Het and KO mice on a No Cys diet with a single dose of 20 mg/kg of anti-GDF15 antibody or control anti-KLH antibody given i.p. 2 hours prior to shifting to a No Cys diet. **B)** Daily food intake in Cse KO animals with anti-GDF15 or anti-KLH antibody.

Given the lack of *Gdf15* Cse DKO mice, we used *Gdf15* KO mice on a No Met No Cys diet to test if even a slight increase in GDF15 (300 pg/ml in B6 mice compared to 800 pg/ml in Cse KO mice on No Cys) could play a role. We observed a significant decrease in weight loss on Day 1 in the *Gdf15* KO compared to B6 WT mice on a No Met No Cys diet. However, the differences did not persist by Day 7, likely due to GCN2 activation compensating for the lack of GDF15, an inverse of what we showed in Reviewer Fig 1 (**new Extended Data Fig 7f and g**).

For FGF21, we could not find any blocking antibody and thus could not perform the recommended experiment. However, Lee et al (See supplemental File 1, Figure 3i) studied Fgf21 Cse double KO mice on cysteine-free diet and observed that these mice lost 7% less weight compared to Cse KO mice, suggesting a clear role for FGF21 in the phenotype.

We now include measurements of serum FGF21 levels on Day 2 (**new Fig 4e, new Extended Data Fig 6e**) to further support our RNA-seq data. Also, we include data from *Fgf21* single KO mice on a No Met No Cys diet to show that FGF21 plays a role in regulating weight loss during cysteine deficiency (**new Extended Data Fig 7h and i**).

3. Another apparent experiment is to assess the importance of ISR by using the well-established ISRIB compound. This experiment would involve examining tissue ISR responses using RNA seq in conjunction with weight loss.

To test the role of ISR in weight loss, we administered the ISR inhibitor (ISRIB) daily to *Cse* KO mice on a No Cys diet (PMID: 33258451). We detected a small but statistically significant decrease in weight loss on Day 1 (**reviewer Fig 3A and B**). However, by Day 7 the ISRIB treated mice lost the same amount of weight as untreated controls (**Reviewer Fig 3A**). ISRIB partially suppressed the phosphorylation of eIF2a (**Reviewer Fig 3C**).

As expected, the levels of GDF15 and FGF21 were downregulated (**Reviewer Fig 3D and E**) (PMID: 34877504), but they were still significantly above the baseline levels found in wt animals (~ 300 pg/ml GDF15 and ~0.2 ng/ml FGF21, **new Extended Data Fig. 7c and e**). These results correlate with previously published data demonstrating that there are *P*-eIF2a-dependent and -independent pathways that converge to upregulate typical ISR transcription in response to sulfur amino acid deficiency (PMID: 27613409, PMID: 28446632). Moreover, a diet low on Cys and Met induces the expression of “classical” ISR genes even in *Atf4*^{-/-} mice (PMID: 33512502). Together, this suggests that compensatory pathway(s) are induced by Cys deprivation, which can trigger an ISR transcriptional signature that is critical for survival on a Cys-free diet.

Reviewer Figure 3: Effect of ISRIB on weight loss. **A)** Weight loss curves of *Cse* KO on No Cys with daily dose 1 mg/kg ISRIB or 2.5 mg/kg ISRIB or vehicle ctrl. **B)** % starting weight across three conditions on Day 1. **C)** % phosphorylation of eIF2a across three conditions in the liver on Day 7. **D)** Serum GDF15 on Day 7. **E)** Serum FGF21 levels on Day 7.

It is essential to emphasize that these experiments do not require the breeding of new mice; instead, they involve straightforward interventions that can provide valuable mechanistic insights into causality. The goal is to determine whether ISR and GDF15 are indeed necessary factors, regardless of the outcome.

ISR is a key cellular response required for survival and adaptation to multiple stresses. It is not surprising that multiple regulatory pathways converge to regulate a similar set of genes defined as ISR. Previous studies in cells and mice have shown that individually, *P*-eIF2a, GCN2, and ATF4 are not absolutely necessary for typical ISR transcription in response to sulfur amino acid deficiency (PMID: 27613409, PMID: 28446632, PMID: 33512502).

To test the role of GCN2 in ISR induction and food aversion, we created a double KO of *Gcn2* and *Cse*. We found that by 8 weeks, nearly all *Gcn2* *Cse* DKO mice developed hindlimb paralysis and had to be euthanized (**Reviewer Video 1**). As both single *Cse* or *Gcn2* KO mice are viable, this strongly indicates that GCN2 is required and activated in *Cse* KO mice to compensate for Cys deficiency. In the absence of the transsulfuration

pathway, GCN2 likely increases the import of dietary Cys, preventing developmental abnormalities. As it has been demonstrated previously that GCN2 can activate ISR, our result provide indirect evidence that ISR is required for adaptation to Cys limitation. Given that GCN2 activation and GDF15 increase can both drive food aversion and compensate for each other, determining the exact contribution of GDF15 and GCN2 is challenging. Additionally, treating mice with BSO alone does not induce the same weight loss as BSO on top of a No Trp diet (**revised Extended Data Fig 4i**). This implies that ISR is needed to induce weight loss.

REFEREE #3

In this manuscript by Varghese et al, the authors found that cysteine deficiency, using a combination of cysteine-deficient diet and transsulfuration pathway-deficient mouse model, induced rapid and reversible weight loss. They meticulously characterized the physiological parameters under cysteine deficiency and found that browning cysteine deficiency causes reduced food intake and rapid browning of adipose tissues. Mechanistically, cysteine deficiency leads to simultaneous activation of the integrated stress response and oxidative stress response and the expression of the anorexic hormone GDF15. Additionally, they found that cysteine deficiency led to the reduction of CoA levels in liver and muscle, causing disruption of metabolism reliant on CoA-containing intermediates.

Collectively, this is a well-designed study that offers a lot of useful insight to the rewiring of systemic metabolism under amino acid depletion. My major concern with this manuscript is its novelty: activation of ISR/OSR by cysteine depletion has been demonstrated in many cellular, physiological and pathological scenarios, and the role of GDF15 in mediating systemic response to ISR is well-documented. Depletion of CoA, a major downstream fate of cysteine, by cysteine depletion, is also expected, and the authors provided limited data on the link between CoA depletion and systemic metabolic effect. Additionally, reliant on Cse-KO animals, though understandably an appropriate tool for inducing cysteine depletion, may nevertheless limit the physiological relevance of the study. Provided that the authors address the specific points below, I would recommend this manuscript for a specialized journal such as Nature Metabolism; for it to be considered by Nature, I would expect other findings to be incorporated into the manuscript that substantially boost its novelty.

We thank the reviewer for the constructive comments and suggestions. We have performed all the feasible experiments and included new data to prove CoA's role in the phenotype, providing more insights into how it rewires metabolism, which we believe has substantially improved the manuscript.

Detailed demonstration of ISR/OSR activation in vivo:

While activation of ISR/OSR by sulfur amino acid (SAA) depletion in cell culture or cancer cell lines was previously reported, it was never demonstrated *in vivo* and in such detail. Our data indicates that the response to systemic cysteine deprivation varies across liver, muscle, and fat, which cannot be deduced from previous studies in cell culture. It was also not known how quickly ISR or OSR would be induced upon cysteine deprivation in an animal, or that the activation of both together would result in phenotypes such as rapid weight loss or ACC degradation.

Met vs. Cys in Sulfur amino acid restriction:

Although increased lifespan and lower fat accumulation/fat loss over weeks of treatment were observed in mice fed on SAA deficient diet, the mechanism and relative importance of Met vs Cys was unknown. Here, we demonstrate that only Cys, not Met, limitation is the primary factor driving rapid fat loss.

Cse KO as a model:

While using a Cse KO is a limitation, it enables us to dissect the effects of cysteine restriction specifically, rather than the combined effect of methionine and cysteine restriction. Using the Cse KO provides insight into the importance of cysteine specifically for weight loss and suggests that CSE is an attractive target for drug development. Working in WT mice has its limitations, as to create cysteine deficiency, we need to restrict both methionine and cysteine. Furthermore, the transsulfuration pathway is able to convert all free and protein breakdown-derived methionine to cysteine in a controlled manner, limiting the cysteine deficiency compared to No Cys in a Cse KO.

Physiological relevance in WT mice:

Nevertheless, to address the physiological relevance concerns, we included an experiment in the revised manuscript where we fed WT (B6) mice equal amounts of control, No Met, or No Met No Cys diets (**new Extended Data Fig 7**). In agreement with our previous experiments, we observed the highest weight loss in mice on the No Met No Cys diet. Furthermore, we detected a statistically significant increase in both GDF15 and FGF21 levels by Day 3 only on the No Met No Cys diet. Using GDF15 and FGF21 single KO mice, we could show that this increase is sufficient to have an affect based on weight loss curves on the No Met No Cys diet. Notably, both hormones were elevated to much lower levels than seen in the Cse KO with No Cys, suggesting that transsulfuration from Met to Cys is important and robust (**new Fig 4d, e, new Extended Data Fig 7**). Additionally, we demonstrated that in WT mice, the No Met No Cys diet induced greater weight loss compared to No Met No Trp (**Fig 1h**), underscoring the importance of Cys for weight loss. Given that Met deficiency induces a specific stress response on top of the response to Cys limitation, dissecting the exact responses in WT mice on the No Met No Cys diet would require substantial work, which we believe is beyond the scope of this paper.

Novel findings on CoA regulation by cysteine:

In the literature, there is no data, even *in vitro*, indicating that cysteine restriction leads to a significant CoA loss. Previous publications demonstrated that CoA is extremely stable, with Vitamin B5 (pantothenate)-deficient diets not leading to appreciable changes in CoA levels even after 2 months (PMID: 3807784). Our work, however, clearly demonstrates that cysteine restriction, unlike B5 deficiency, results in rapid CoA depletion. Our results have several important and novel implications:

1. Cysteine tightly regulates CoA levels.
2. There is significant turnover of CoA.
3. CoA degradation to B5 and then resynthesis provides a means of interorgan transport of CoA.
4. Cysteine deficiency induces an attempt to redistribute CoA between tissues.

Linking CoA depletion to metabolic effects:

In the revised manuscript, we have included more data, linking CoA depletion to systemic metabolic effects and showed direct evidence for it. We included new experiments, including ¹³C glucose tracing suggested by you, that conclusively demonstrate that CoA depletion specifically mediates the metabolic rewiring (**new Extended Data Fig 10**). We also included new data in Figure 5 to conclusively show a role for CoA in regulating weight dynamics (**new Fig 5k-n**).

Comparative analysis between Cse KO and Cse Hets:

The physiological differences between Cse KO and Cse Hets are limited, as we see very limited differences in gene expression in the liver and muscle in mice on the control diet, showing that the Cse KO is different from the Het only during cysteine limitation. Data from the No Trp+BSO experiment also show that creating both ISR and OSR simultaneously will result in substantial weight loss, even in the Cse Hets (**revised Fig 4i**).

It seems counterintuitive that cysteine depletion leads to substantially decrease in Coenzyme A and switch towards more utilization of lipid, as shown by the RER. Could the authors provide an explanation for this?

This is a very good observation, and we thank the reviewer for pointing this out. There is previous literature (PMID: 2087902, 5043848) showing that the Km for acyl-CoA synthetases is lower than that of pyruvate dehydrogenase. This would make lipid the preferred substrate for metabolism when CoA is limited, leading to the reduced RER. Also, mice switched to utilization of lipids almost immediately when shifted to a No Cys diet (**revised Fig. 2d**) and burned almost completely through their lipid stores by Day 7 (**Fig. 2f**), while CoA decreased gradually (**Fig. 5b and d**).

The rapid browning of white fat in 1H is quite striking. Is there data on total energy expenditure that quantitatively measures the contribution of browning on weight loss?

When we normalize peak energy expenditure to body weight, we observe increased energy expenditure per unit body weight, suggesting that increased browning contributes to additional weight loss (**new Extended Data Fig 10j**). However, given that there is increased dependence on glycolysis in the Cse KO mice as shown in Fig 5 and that energy expenditure is measured based on CO₂ released, we may be underestimating the difference.

Additionally, as this reviewer suggested, we performed heavy glucose tracing experiments and found evidence to indicate that futile creatine cycling might play a significant role in the increased thermogenesis (see below). Specifically, heavy glucose tracing experiments demonstrated tracer flow into guanidoacetate and creatine (**new Extended Data Fig 10b, e-h**). Accordingly, we detected an increase in creatine in the liver at Day 6 of *Cse* KO mice on a No Cys diet (**new Extended Data Fig 10k**).

As Cys deficiency progresses, CoA concentration decreases, and mitochondria activity diminishes (**Fig. 5**). That would require an alternative for UCP1-dependent thermogenesis. Indeed, in adipose tissue, we detected a strong upregulation of the cytoplasmic creatine kinase, M-type (*Ckm*), while the mitochondrial creatine kinase (*Ckmt1*), was significantly downregulated (**new Extended Data Fig 10i**). Together, these results suggest that in the later days on a Cys-deficient diet, *Cse* KO mice rely on futile creatine cycling in the cytoplasm for thermogenesis.

Data from Lee et al (Supplemental file 1, Fig 2f, 3c, d) also suggest an increase in energy expenditure and an increase in temperature in brown adipose tissue due to increased browning. Their data further indicate this effect is driven by FGF21 (Supplemental file 1, Fig 3j).

The author implies that cysteine loss, compared to tryptophan loss, leads to more activated ISR due to its low levels and high demands. It would be very helpful if the author could justify this claim by measuring the decrease in cysteine/tryptophan upon dietary depletion (for measuring cysteine this may require derivatization). The levels of activation of ISR is also related to the K_m of the respective aminoacyl-tRNA synthetase; the authors could also consider discussing this in their manuscript.

We thank the reviewer for pointing out this possibility. To determine which amino acid is depleted more quickly, we measured Cys and Trp levels after one day on control, No Cys, or No Trp diets. Indeed, Cys was depleted faster on the No Cys diet compared to Trp on the No Trp diet (**Reviewer Fig 4**). After one day, Cys levels dropped 53% in *Cse* KO and 6% in *Cse* Het on the No Cys diet compared to the control diet, while Trp levels decreased by ~14% in both strains on the No Trp diet. Thus, the higher ISR response can be explained by the relatively faster depletion of Cys compared to Trp (**Reviewer Fig 4**).

Reviewer Figure 4: Tryptophan and cysteine levels after 1 day on the diet without either. Levels of **A) Tryptophan** and **B) Cysteine** in the liver of Cse Het and KO after 1 day on a diet deficient in each respectively. The data is normalized to the level in Het or KO with the control diet.

For the second part, we could only find Km values for Cys and Trp aminoacyl-tRNA synthetase (AARS) from *E coli*, which were similar at 5×10^5 for Cys and 3×10^5 for Trp (PMID: 371674, 4869215). Given the lack of data in mice, we believe it would be difficult to discuss the contribution of differences in AARS to the induction of ISR.

Compared to NoTry+BSO, NoCys diet seems to induce more weight loss only after 5 days. However, most transcriptomics/metabolomics measurements were performed on day 2. Could the authors provide a justification for choosing this time point? BSO is a relatively weak inhibitor and prolonged treatment may lead to a compensatory increase in the expression of genes in GSH synthesis such as SLC7A11, GCLC, GCLM, which may dampen the effect of drug treatment at the later time points.

We understand the reviewer's point, however, we believe that transcription changes on Day 2 better represent the response to No Cys or No Trp+BSO diets. We specifically intended to see the early changes in gene expression while the fat is not completely depleted, to identify similarities and differences between these two conditions. We found that upregulation of OSR and downregulation of FA and cholesterol biosynthesis are very similar between these two treatments in the liver (**revised Fig. 4j and k**). However, ISR upregulation by No Trp+BSO is weaker compared to the No Cys diet (despite similar $P\text{-eIF2}\alpha$), suggesting that another factor (faster Cys vs. Trp depletion and CoA deficiency) boosts the ISR (**revised Fig. 4j**).

We respectfully disagree with the assessment that BSO is a weak inhibitor, given that BSO treatment in fact reduced the GSH faster than No Cys as we showed in Fig 4m. Furthermore, the addition of BSO to No Cys in a Cse KO led to even higher weight loss than No Cys alone on Day 1, suggesting BSO is much more efficient than Cys limitation

in reducing GSH content. To further verify BSO efficiency we now determined GSH concentration on Day 7 (**Reviewer Fig 5**). We did not detect any statistically significant increase in GSH in mice livers on the No Trp+BSO diet compared to the No Cys diet. This result argues that although *Gclc* expression was upregulated (**revised Fig. 4j**), BSO still efficiently inhibits GSH synthesis in our model.

Reviewer Figure 5: GSH levels in the liver after 7 days on the No Cysteine or No Tryptophan + BSO diet in Cse KO mice

In characterizing the metabolic inefficiency caused by CoA deficiency, the authors used many metabolite levels in tissue versus urine as a proxy for metabolic flux; this is a very indirect snapshot of the metabolic flux. If possible, the author could consider in vivo tracing with isotope-labeled metabolite to have better quantification of metabolic inefficiency.

As suggested by the reviewer, we performed ^{13}C glucose tracing on Day 3 of the No Cys diet in *Cse* Het and KO mice (**new Extended Data Fig 10**). Mice were given a single bolus of ^{13}C glucose orally at 1.25 mg/kg after 18 hours from the last meal, and liver (45 minutes and 2 hours post-gavage) and urine (2 hours post-gavage) samples were collected for metabolomics analysis. We could not use continuous ^{13}C glucose infusion, as significant weight loss in *Cse* KO on the No Cys diet would interfere with a surgically implanted tubing. Since we saw very limited labeled molecules in the liver at the 2-hour time point, we decided to focus only on the 45-minute time point.

Reduction in CoA affects multiple metabolic pathways such as carbohydrate metabolism, amino acid metabolism (especially given the increase due to inhibition of protein synthesis), lipid metabolism etc. To determine how CoA reduction could affect each pathway would require multiple tracing experiments using multiple ^{13}C -labeled

amino acids or lipids, which would be beyond the scope of this paper but would definitely serve as a great follow up for this story.

The glucose tracing data revealed both expected and new aspects of how CoA deficiency rewires glucose metabolism:

1) **Lactic acid**: All four of the Cse KO mice at 45 minutes displayed high levels of U-¹³C-lactic acid (m+3), while its level was undetectable in three out of four Cse Hets, consistent with higher production of lactate from pyruvate and diversion of pyruvate from the TCA cycle (**new Extended Data Fig 10b, c**).

2) **Creatine**: We saw a significant increase in m+2 ¹³C-creatine in both urine and liver of the Cse KO mice, as well as a significant increase in heavy guanidoacetate, the precursor of creatine (**new Extended Data Fig 10d-h**). Creatine is synthesized from arginine and glycine; the latter is derived from serine, which is in turn made from 3-phosphoglycerate, a lower glycolytic intermediate. Increased creatine synthesis, which generally occurs in the liver, is consistent with the need to maintain an increased creatine and phosphocreatine pool that can buffer ATP concentrations and sustain thermogenesis.

3) **Orotate**: Limited CoA concentration would inhibit acetyl-CoA biosynthesis, leading to pyruvate being carboxylated to form oxaloacetate, which is then transaminated to generate aspartate. Aspartate is the precursor of orotate, a critical intermediate in pyrimidine synthesis. We observed a significant increase in m+3 ¹³C-orotate in the urine, revealing another mechanism by which mice can waste glucose carbon in the cysteine-deprived state (**new Extended Data Fig 10b, i and j**).

Additionally, the contribution of metabolic inefficiency to weight loss is unclear. (How) Does it contribute to increased lipolysis in the adipose tissue? How much energy is lost via excretion as urinal metabolites?

This is another important point raised by the reviewer. Based on the rapid loss of fat occurring before the substantial loss of CoA, it is likely that increased lipolysis starts independently and is likely initiated by food aversion. Later on, metabolic inefficiency may contribute to a further increase in lipolysis, but it would be hard to determine the exact amount of lipolysis mediated by metabolic inefficiency versus browning versus GDF15 induced lipolysis, etc.

The second point is also an important question. We agree that we only provide a snapshot of what is happening with the urine data we have shown. Given the differences with RER that we see with circadian rhythm, it is likely that different metabolites would be found in urine if we collected it at different times. To know how much energy is lost in urine, we would need to collect urine continuously throughout the day, which is currently not feasible.

Cysteine affects OXPHOS via multiple pathways (such as CoA, GSH and Fe-S cluster synthesis), so defective OXPHOS is very weak evidence that CoA is mediating the effect of cysteine depletion. The authors should try to clarify the contribution of CoA in mediating OXPHOS defects.

This is an extremely important observation and question regarding the significance of the CoA contribution that we propose in our paper.

To determine the role of CoA, we relied on the fact that during the phase of No Cys, the mice also lost pantothenic acid (vitamin B5) in urine since B5 is a water-soluble vitamin that can't accumulate, and CoA resynthesis is limited due to lack of cysteine (**Fig 5i, j**). Thus, at Day 7, along with the depletion of other cysteine containing molecules, the mice would also have reduced B5 levels across tissues. Consequently, if we gave mice, after 7 days of No Cys, a diet with No B5 but with cysteine, we could fully rescue all cysteine-containing molecules except CoA, which would be limited to the remaining tissue B5 levels or what was absorbed that day (**new Fig 5k**).

To verify this hypothesis, we reverted *Cse*^{-/-} mice on Cys-free diet to cysteine-sufficient but B5-deficient diet. As expected, the mice failed to gain as much weight as on the control, B5- and cysteine-sufficient diet (**new Fig. 5l**). While we detected the GSH rescue, the CoA levels were still significantly lower on the B5-deficient diet (**new Fig 5m, n and new Extended Data Fig 13a, b**). Furthermore, giving B5 in drinking water after 7 days of the B5-deficient diet readily rescued the difference in weight recovery (**new Extended Data Fig 13c**).

Taken together, our experiments clearly demonstrate that:

- 1) CoA is quickly and progressively lost on a Cys-free diet
- 2) Cys, but not B5 availability, is a key determinant of cellular CoA levels
- 3) CoA loss shifts cells to anaerobic glycolysis and accelerates the wasting of intermediary metabolites in the urine, decreases mitochondrial utilization, and reduces TG production.

These collectively contribute to quick weight loss in *Cse*^{-/-} mice on a Cys-free diet.

Direct rescue of CoA is not possible because cells cannot directly take up CoA and would degrade it to pantothenic acid and cysteamine. Pantothenic acid would then be transported into the cell via SMVT, but given the low cysteine in the No Cysteine *Cse* KO condition, there will not be an increase in CoA level.

Inhibition of *Vnn1* is also not possible as there is only a rat- and human-specific inhibitor and not a mouse-specific one (PMID: 23270378).

Referee #1 (Remarks to the Author):

I have reviewed the response to reviewers document and the revised manuscript. While the authors have addressed some of my concerns with respect to mechanisms I do not believe they have adequately addressed my concern about the very limited sample sizes for many experiments which lay the foundation for the paper. Specifically, it is stated in the response to reviewers that "All experiments presented in the revised manuscript, except for the metabolomics and sequencing, were conducted at least twice with a minimum of three replicates per group". From my understanding of the above statement then there should be a minimum of a $n=6$ per group, which would seem a reasonable number of mice to avoid a type 2 error and ensure reproducibility. However, from data presented in Figures 1-2 and Extended Data Figures 1-3 it is clear this is not the case as most experiments have an $n=2-4$. Moreover the sample size in many of the new experiments are also below this threshold (i.e. figure 4,5,6, extended data figure 6a, b and 7 all $n=3-4$).

We thank the reviewer for insightful comments across both rounds of reviews, which helped us strengthen the mechanistic aspects of our paper and enhance reliability of our results.

We acknowledge that, in an ideal scenario, a higher 'n' would further bolster our claims and enhance reproducibility. However, we believe that our approach - combining multiple forms of evidence to support all major points in the paper, alongside the consistent weight loss curves and other metrics across various experiment models - demonstrate high robustness.

That said, we have included previously unpooled repeats to increase the 'n' in multiple experiments and performed additional repeats to further support our conclusions. Initially, we did not pool all data for the following reasons:

1. We performed many experiments using mice of both biological sexes, and as shown in Extended Data Figure 1k, there is a noticeable difference in weight loss curves between male and female mice. Aggregating all the repeat data would introduce noise and could obscure proper interpretation of both the weight curves and metabolic cage data.
2. While weight measurements were primarily taken at 24-hour intervals, some repeats included time points taken at 18-hour or 30-hour intervals due to scheduling constraints. Pooling these data would introduce further noise, making interpretation more challenging.
3. Differences in starting weights led to variations in baseline food consumption across repeats, which would further increase noise in the aggregated data.

For the metabolic cage experiments shown in Figure 2a-d, we have added 3 new extended data figures (Extended Data Fig 2a-c) with the profiles in female mice. We opted not to combine the data because doing so would make interpretation difficult, due to differences

in exact point of time at which they were fed. Presenting the data separately clearly illustrates that the effect is reproducible. Other updated graphs include Fig: Fig 1e, Fig 4a, Fig 5c, d among the main figures and Fig 1l, 1p, 1q, 4a, 4b, 5f, 5g and 8h among extended data figures.

We also apologize for the misstatement in our previous submission "All experiments presented in the revised manuscript, except for the metabolomics and sequencing, were conducted at least twice with a minimum of three replicates per group". We neglected to include word "total", and the correct statement should read: "at least twice with a **total** minimum of three replicates per group". While we acknowledge that a higher number would be ideal, the reproducibility of our findings across different types of experiments mitigates this concern.

Moreover, the fact that another lab was able to reproduce many critical aspects of our study using a different Cse KO model and a different cysteine-free diet in a separate facility further underscores the high reproducibility of our findings.

Referee #2 (Remarks to the Author):

The authors have done a good job of responding.

Thank you for your kind assessment. Your comments have greatly strengthened and improved our manuscript.

Referee #3 (Remarks to the Author):

The authors have added evidence that significantly strengthened the observations in the original submission. In particular I'm pleased to see the inclusion of the B5 diet experiment to pinpoint the role of CoA in mediating weight loss as well as tracing with labeled glucose to better characterize the metabolic inefficiency caused by CoA depletion.

Overall the revised manuscript is much more comprehensive in characterizing the metabolic phenotypes and offered more mechanistic explanations. However the how (much) exactly ISR/OSR and "metabolic efficiency" contribute to rapid weight loss still lacks convincing explanation, possibly due to limitations in currently available technologies.

Thank you for your kind assessment. We agree that it would be valuable to determine how much each response contributes to weight loss, and we hope to develop or apply new techniques in the future to address this as part of a follow-up study. Once again, we appreciate your comments, which have helped us to strengthen our study and improve our manuscript.

Response to Reviewer 1

We certainly agree that experiments should have sufficiently high 'n' to minimize Type II errors. However, Type II errors are influenced not only by the sample size but also the magnitude of differences and the variability within each group (Cohen's d).

Good scientific practices recommend achieving a statistical power of at least 0.8 (1-probability of a Type II error) to ensure confidence in the results. To address Reviewer 1's concern, we have now included a table (**New Supplementary Table 7_Power Analysis**) outlining the power calculations for each experiment where we claim statistical significance, along with the corresponding sample sizes required to achieve a power of 0.8. We have also added more repeats to both **Figs 5m** and **5n** in the amended manuscript to increase power.

The data demonstrate the following:

Main Figures:

- For 21 of the 25 main figure panels where power analysis is applicable, the statistical power is at least 0.8. This demonstrates that our sample sizes ($n \geq 3$) are sufficient to support all major findings in the manuscript.
- For the remaining 4 panels that do not meet a power of 0.8, we provide additional evidence to substantiate our claims (detailed below). This includes independent experiments yielding consistent result.
- Moreover, we integrate multiple lines of evidence - such as RNA-seq data, Western blot analysis, hormone measurements, and various methods to measure metabolic inefficiency – and present consistent weight loss curves and related metrics across different experimental models. These collectively strengthen the validity of our conclusions.

Extended Data Figures:

- For 17 of the 22 extended data figure panels where power analysis is applicable, the statistical power is at least 0.8.
- Of the remaining 5 panels, 4 have corroborating data from related molecules or repeat experiments conducted on alternate days, which provide additional confidence, as detailed below.
- Thus, even for the extended data, we believe our sample sizes ($n \geq 3$) are adequate for most additional claims made in the manuscript.

Furthermore, we note that several recently published Nature papers in the field of metabolism include figures where specific panels use sample sizes of $n = 3$, similar to our approach.

<https://www.nature.com/articles/s41586-024-08379-9> (Figure 3),

<https://www.nature.com/articles/s41586-024-08317-9> (Figure 1, 2 and 5),

<https://www.nature.com/articles/s41586-024-08335-7> (Figure 1 and 2),

<https://www.nature.com/articles/s41586-024-08329-5> (Figure 1-3),

<https://www.nature.com/articles/s41586-024-08348-2> (Figure 1-3)

These examples further support the sufficiency of our sample sizes (see below), particularly in the context of the large effect sizes and low variability observed in our experiments.

Moreover, we provide detailed evidence below showing that, for all major claims, our analysis has sufficient statistical power, ensuring that the overall conclusions are robust and well-founded.

However, if any of the reviewers or the editor feel that a sample size should be justifiably increased for any specific experiment, we are willing to do so.

We hope this response sufficiently addresses Reviewer 1's concern.

Power analysis for experiments supporting major claims:

- 1) Cysteine deficiency induces rapid weight loss: **Fig 1b and 1c - Power (P)=1, n=5.**
- 2) Cysteine deficiency induces the largest weight loss compared to other essential amino acids: **Fig 1b, Extended Data Fig 1b, c - P (No Iso to No Cys in KO) =0.89; for all other comparisons with No Cys in KO - P=1, n≥4.**
- 3) Cysteine deficiency induces food aversion: **Fig 1e - P= 0.99, n=6.**
- 4) Cysteine deficiency induces food aversion-independent weight loss: **Fig 1f - P=0.99, n≥3.**
- 5) Methionine-cysteine dual deficiency in B6 mice induces greater weight loss than methionine-tryptophan dual deficiency: **Fig 1g - P=1, n≥4.**
- 6) Cysteine deficiency induces increased fat utilization (by RER): **Fig 2d - P=0.99; ED Fig 2c - P=0.88, n≥3** in each repeat per strain.
- 7) Cysteine deficiency induces rapid fat mass loss (by DEXA), **Fig 2e - P=0.99** for both Ad libitum and Calorie Restriction comparisons, **n=4 in each group.**
- 8) Cysteine deficiency induces rapid GSH loss, **Fig 4a - P=1 for both liver and muscle, n≥7.**
- 9) Cysteine deficiency induces ISR: **Fig 4c - P=0.99, n≥5.**
- 10) Cysteine deficiency induces rapid ACC1 loss: **Fig 4f - P=1, n=4.**

- 11) Cysteine deficiency-induced weight loss is not fully explained by oxidative stress and ISR: **Fig 4i - P=1, n=4.**
- 12) Cysteine deficiency induces CoA loss:
Fig 5b - Liver (Day 2) P=0.91, n=4.
Fig 5c - Muscle (Day 2) P=0.61; Liver (Day 7) P=1, n≥7.
- 13) Cysteine deficiency forces dependence on glycolysis: **Fig 5g - P=1, n=4.**
- 14) Lack of B5 in recovery diet containing cysteine limits weight regain: **Fig 5l - P=1, n=5.**
- 15) Lack of B5 in recovery diet containing cysteine limits CoA recovery:
Fig 5m – Liver - P=0.91, n≥8; Muscle - P= 0.67, n≥8. For muscle, two independent repeats show statistically significant differences.

For RNA-seq conclusions, we used n=8 for Het No Cys and n=7 for KO No Cys in the liver, the major groups for our comparisons and claims.

Justification for experiments with Power below 0.8:

Main figures

- 1) **Fig 4d: GDF15 serum measurements (P=0.78, n≥7).**

This figure derives from two experiments, both showing higher GDF15 in the KO No Cys group. The first experiment yielded a p-value of 0.057, and the second a p-value of 0.015 (Reviewer Figure 1). Moreover, RNA-seq data from the liver (ED Fig 3b, Supplementary Table 2) clearly show GDF15 induction.

Reviewer Figure 1: GDF15 serum level on Day 2 of No Cys in Het and KO mice in individual repeats.

2) **Fig 4h: Serum triglyceride measurements (P=0.67, n≥5).**

ED Fig 6E (Day 7) demonstrates a similar decrease in serum triglycerides in the KO EGFP group compared to the Het EGFP group. Supporting evidence includes a clear decrease in serum free fatty acids on Day 2 (Fig 4g) and a drop in ACC1 levels in the liver on Day 2 (Fig 4f).

3) **Fig 5c: Muscle CoA levels on Day 2 (P=0.61, n≥7).**

While the liver shows a clear decrease in CoA, the difference in muscle is smaller necessitating an $n > 10$ for sufficient power. The upregulation of *Vnn1*, which degrades CoA in muscle (Day 2), further corroborates the observed decrease.

4) **Fig 5n: Muscle CoA levels 7 days after recovery (P=0.67, n≥8).**

Data are derived from two experiments, both showing lower CoA levels in the No Cys to No B5 group compared to the No Cys to Ctrl group. The first experiment yielded a p-value of 0.048, and the second a p-value of 0.007 (Reviewer Figure 2). Higher variation in the control recovery group in the first experiment caused the lower power.

Reviewer Figure 2: Muscle CoA in KO mice following either Ctrl diet with or without Pantothenic acid after 7 days of no cysteine in individual repeats.

Extended Data Figures

1) **ED Fig 5b: Serum GDF15 measurement on Day 3 in B6 mice (P=0.635, n=4).**

GDF15 measurements on Day 7 (ED Fig 5c), demonstrate the same statistically significant increase in the No Met No Cys group compared to the No Met group, with lower variation and sufficient power. Also, GDF15 single KO shows reduced weight loss on Day 2 in the No Met No Cys compared to controls (Fig 5h), further validating the data.

- 2) **ED Fig 5e: Serum FGF21 measurement on Day 7 in B6 mice (P=0.435, n=4).**
Measurements on Day 3 (ED Fig 5d) reveal a statistically significant increase in the No Met No Cys group compared to the No Met group, with lower variation and sufficient power.
- 3) **ED Fig 6e: Serum triglyceride measurements (P=0.716, n=5).**
Data in Fig 4h (Day 2) show a similar decrease in serum triglycerides in the KO No Cys group compared to the Het No Cys group. This finding is supported by a clear decrease in serum free fatty acids on Day 7 (ED Fig 6d).
- 4) **ED Fig 8d, Orotate M+3 levels in urine from glucose tracing experiment (P=0.77, n≥4).**
Data in Fig 5e show a consistent increase in orotate levels in both liver and urine, suggesting increased rerouting of metabolites to orotate.
- 5) **ED Fig 8j: Normalized energy expenditure (EE) in metabolic cages (P=0.65, n≥7).**
High variation in this figure resulted in lower power. To address this, we can calculate average EE over a 2-hour period (reducing variation), repeat the experiment to increase n , or exclude this figure, as the editor or reviewers deem necessary.